# TRANSFORMERS STRUGGLE TO LEARN TO SEARCH

**Abulhair Saparov**[×,1]     **Srushti Pawar**[2]     **Shreyas Pimpalgaonkar**[2]     **Nitish Joshi**[2]
**Richard Yuanzhe Pang**[2]        **Vishakh Padmakumar**[2]       **Seyed Mehran Kazemi**[3]
**Najoung Kim**[*,4]       **He He**[*,2]
[1]Purdue University, [2]New York University, [3]Google, [4]Boston University
`asaparov@purdue.edu`

## ABSTRACT

Search is an ability foundational in many important tasks, and recent studies have shown that large language models (LLMs) struggle to perform search robustly. It is unknown whether this inability is due to a lack of data, insufficient model parameters, or fundamental limitations of the transformer architecture. In this work, we use the foundational graph connectivity problem as a testbed to generate effectively limitless high-coverage data to train small transformers and test whether they can learn to perform search. We find that, when given the right training distribution, the transformer is able to learn to search.

We analyze the algorithm that the transformer has learned through a novel mechanistic interpretability technique that enables us to extract the computation graph from the trained model. We find that transformers perform search at every vertex *in parallel*: For each vertex in the input graph, transformers compute the set of vertices reachable from that vertex. Each layer then progressively expands these sets, allowing the model to search over a number of vertices exponential in $n_{\text{layers}}$.

However, we find that as the input graph size increases, the transformer has greater difficulty in learning the task. This difficulty is not resolved even as the number of parameters is increased, suggesting that increasing model scale will not lead to robust search abilities. We also find that performing search in-context (i.e., chain-of-thought) does not resolve this inability to learn to search on larger graphs.

## 1 INTRODUCTION

The ability to search is fundamental in many important tasks, such as reasoning (Yao et al., 2024; Kazemi et al., 2023; Hao et al., 2023), planning (Stein & Koller, 2023; Valmeekam et al., 2022), and navigation (Ding et al., 2024). Recent work has demonstrated that transformer-based large language models (LLMs) struggle with proof search (Saparov & He, 2022; Valmeekam et al., 2022; Kambhampati et al., 2024). It is unknown whether this shortcoming is due to a lack of data, insufficient model parameters, or a fundamental limitation of the transformer architecture. In any case, as the scale of LLMs continues to increase, it is yet unclear whether future LLMs, equipped with more data, parameters, and compute, will be able to perform search and planning. Chain-of-thought and similar prompting techniques (Wei et al., 2022b; Nye et al., 2021) have enabled LLMs to decompose the search task into the repeated task of predicting the next step along the path to the goal. However, even in this setting, in the worst case, in order to avoid making a "wrong turn," the model must perform the search *within* its forward pass to determine the correct next step. And LLMs have been observed producing errors or hallucinations after taking such a wrong turn (Saparov & He, 2022).

We aim to shed light on this question by training small transformer models on a simple yet foundational search task: Given a directed acyclic graph (DAG), a start vertex, and a goal vertex, find the next vertex along a path from the start to the goal vertex. This task is the backbone of many reasoning problems as it is equivalent to proof search in a simplified logic which is a subset of almost any logic: The model must solve this task if there is any chance to generalize to more complex search and reasoning tasks. We demonstrate experimentally that transformers can indeed be taught to search, when given the right training distribution. The training distribution must be carefully constructed so as to preclude the usefulness of shortcuts or heuristics that would otherwise prevent the transformer from learning a robust and generalizable search algorithm. By automatically generating such examples, we provide the transformer with effectively limitless and idealized training data, with which we can estimate an "upper bound" on the transformer's ability to learn to search.

---

*Equal contribution. [×]Much of the work was completed while author was at New York University.

When the model *does* learn the constructed training distribution (i.e., reaches 100% training accuracy), it is able to correctly perform search in almost any graph. We aim to determine the algorithm that the model has learned to solve the search task, and to measure the extent to which the model uses such an algorithm or relies on heuristics. We develop a mechanistic interpretability analysis to study the learned algorithm. We find that transformers perform search simultaneously on all vertices of the input graph, where for each vertex, the transformer stores the set of vertices reachable within a certain number of steps. Each layer successively extends the sets of reachable vertices, thereby allowing the model to search over a number of vertices exponential in the number of layers.

However, we find that as the input graph size increases, the transformer has increasing difficulty in learning the training distribution. We find that increasing model scale does not alleviate this difficulty, suggesting that simply increasing the size of the transformer will not lead to the robust acquisition of searching and planning abilities.

We also consider modified versions of the search task where the model is permitted to output intermediate tokens (akin to chain-of-thought prompting; Kojima et al., 2022; Wei et al., 2022b). More specifically, we test depth-first search and (zero-shot) selection-inference prompting (Creswell et al., 2023). We find that while it is easier to teach the model to solve this task, requiring a constant number of layers, the model still struggles on larger input graphs.

Thus, our results suggest that future transformer-based models will not solve the search task with standard training, and alternative training approaches may be necessary.[1]

## 2 RELATED WORK

**Search abilities of transformers.** A number of studies have explored the search capabilities of transformers and LLMs. Benchmarks have shown that LLMs can perform some graph reasoning tasks and related tasks such as repeated function composition and the $k$-hop induction head task (Fan et al., 2024; Fu et al., 2024; Sanford et al., 2024a;b), but they are limited to small graphs, relative to graphs we consider. Ruoss et al. (2024); Gandhi et al. (2024); Shah et al. (2024) find that a transformer can learn to approximate or simulate a search algorithm, but with a performance gap, and they do not test whether this gap is lessened by increasing model scale or training. Wang et al. (2023); Bachmann & Nagarajan (2024) show that LLMs can do some graph reasoning but are fooled by spurious correlations. Similarly Zhang et al. (2023) find that transformers are unable to learn to perform proof search since they strongly prefer heuristics. In this work, we also show that transformers are highly sensitive to the training distribution, but if extra care is taken to remove heuristics, they are able to learn to search. Zhang et al. (2024) show that LLMs struggle on real-world graph reasoning tasks. Borazjanizadeh et al. (2024) find that LLMs have difficulty on diverse search problems. However, this is in contrast with work on the theoretical expressiveness of transformers. Merrill & Sabharwal (2024) show that with chain-of-thought, transformers can simulate any Turing machine. However, their results do not indicate whether it is possible to *train* a transformer to perform any task. In fact, we find that even if the transformer is permitted to generate intermediate tokens, it is challenging to learn to search on larger graphs. Sanford et al. (2024a;b) show that transformers need a logarithmic number of layers to perform the graph connectivity task, which is supported by our identification of the algorithm that transformers acquire during training.

**Mechanistic interpretability.** There is a large amount of work that seek an algorithmic understanding of transformers trained on various tasks. Hou et al. (2023); Kim et al. (2024) look for evidence of specific circuits/algorithms in the transformer's activations and attention patterns. In our approach, we do not require the algorithm be known a priori. Rather, we reconstruct the computation graph from the model activations and attention patterns. We make heavy use of *activation patching* (Vig et al., 2020b; Geiger et al., 2021; Heimersheim & Nanda, 2024). Brinkmann et al. (2024); Kim et al. (2024); Stolfo et al. (2023) apply mechanistic interpretability analysis to better understand transformer behavior in reasoning, and Yang et al. (2024); Jenner et al. (2024) present evidence that LLMs perform *shallow* searches during the forward pass. Ivanitskiy et al. (2023) train small transformer models on a maze search task and find internal representations of the maze map.

**Scaling laws.** Scaling laws are empirically-supported hypotheses about the long-term behavior of machine learning models on a task, as a function of the model size, the amount of data, and compute (Kaplan et al., 2020; Henighan et al., 2020; Hoffmann et al., 2022). Caballero et al. (2023) explore scaling laws on a wide multitude of tasks, but not including search, reasoning, or planning. Wei

---

[1]All code for generating data, training and evaluation is open-source and freely available at `github.com/asaparov/learning_to_search`.

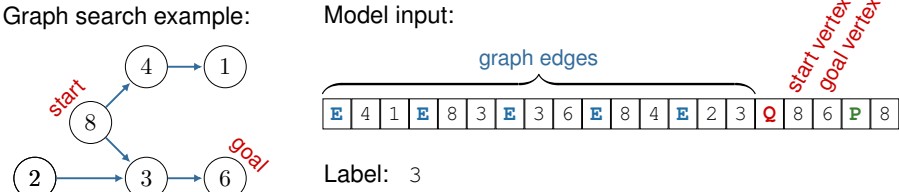

FIGURE 1: **(top left)** Example of a search example on a directed acyclic graph and **(top right)** the corresponding transformer input and output label. **(bottom)** An equivalent proof search problem in implicational propositional logic, rendered in natural language.

et al. (2022a) posit that there may exist phase transitions as scale increases, where certain abilities "emerge." However, Schaeffer et al. (2023) find that this emergent behavior can be explained by careful selection of metrics. Du et al. (2024) argue that training loss is a better measure of model ability, independent of scale.

## 3 SEARCH IN DIRECTED ACYCLIC GRAPHS

In order to test whether transformers can learn to perform search, we need a way to produce a large number of search problems on which the model can be trained and evaluated. We do so by generating search problems on directed acyclic graphs (DAGs). Each example consists of a DAG, a start vertex, and a goal vertex (which is guaranteed to be reachable from the start vertex). The model's task is to output the next vertex on the path from the start to the goal vertex. We characterize the difficulty of an example by its *lookahead*: the number of steps that the model must search from the start vertex to find the goal. More precisely, for any example graph, let $P$ be the path from the start to the goal vertex, and $S_i$ be the paths from the start vertex that are otherwise disjoint with $P$, then the lookahead $L$ is $min\{|P|, \max_i |S_i|\}$. An example graph, along with the corresponding transformer input, is shown in Figure 1. In this graph, the lookahead is 2.

We experiment with three distributions of DAG search problems: (1) two simple distributions where we pay no special attention to heuristics, which we call the "naïve" and "star" distributions, and (2) a more carefully constructed distribution where we take care to prevent the model from exploiting heuristics to solve the task, called the "balanced distribution."

**Naïve distribution.** We adapt the Erdős–Rényi random graph distribution (Erdös & Rényi, 1959) to generate directed acyclic graphs: We arrange a set of vertices in linear order from left to right (i.e., topological order) and randomly sample edges between pairs of vertices. To ensure there are no cycles, all edges are oriented to the right. To reduce the density of the graphs, we limit the in-degree of any vertex to 4, since the lookahead is typically very small in dense graphs. See Section A.1.1 for details on the generative process.

**Balanced distribution.** While easy to describe and implement, the naïve distribution strongly tends to generate graphs where the lookahead is very small (typically $L = 1$ or 2; see Figure 8). In fact, it becomes exponentially less likely to generate examples with greater lookaheads. In order to both train and evaluate the model's ability to search multiple steps to find the goal vertex, we need an efficient way to generate examples where the model is required to search for additional steps in order to find the goal. In addition, to prevent the model from relying on heuristics to perform search, we must take care to ensure that heuristics are not useful to solve the search problems in the training distribution. Thus, we design the balanced distribution to specifically generate graphs with a lookahead parameter $L$ (see Section A.1.2 for details), which we use to produce training data where the lookahead is uniformly distributed.

**Star distribution.** We experiment with an additional distribution over graphs where the vertices are arranged in a star-shape (see Fig 1 in Bachmann & Nagarajan, 2024). The vertex at the center is the start vertex, and there are $k$ "spokes" that radiate outwards from the center, where each spoke is a linear chain of $L$ vertices. The goal vertex is at the end of one of the spokes.

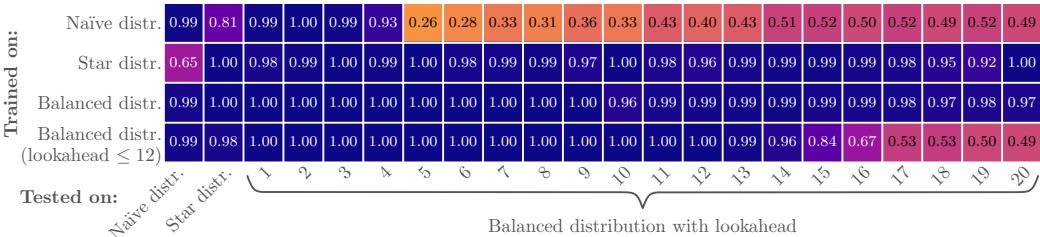

FIGURE 2: Accuracy of model with a maximum input graph size of 41 vertices trained on 883M examples from the naïve distribution, vs star distribution, vs the balanced distribution with lookaheads $L \leq 20$ (which is the maximum for the input size), vs the balanced distribution with $L \leq 12$ for the last row. All evaluation is on held-out examples.

## 3.1 EXPERIMENTS

We train transformer models, with the same architecture as GPT-2 (Radford et al., 2019) with ReLU activation. In order to facilitate mechanistic interpretation of the trained model behavior, we use 1-hot embeddings for both the token and the absolute positional embedding. Furthermore, the token embedding and positional embeddings are concatenated rather than added to form the transformer input. We use 1 attention head per layer. The feed-forward dimension is the same as the model dimension. Since the edges in our inputs are randomly ordered, it would help for each token to be able to attend to any other token, rather than only the preceding tokens. As such, the model does not use a causal mask when computing attention. We train an 6-layer model with hidden dimension 16.

To simulate training on samples of a very large corpus of data, we utilize *streaming training*. We continually sample batches from the generative process throughout training, instead of sampling batches from a fixed training set. The first few batches are reserved for testing, and subsequent batches are filtered via exact matching to remove any examples that appear in the test set, to ensure that the examples in the test set are unseen. In all our experiments, we use the `Sophia` optimization algorithm (Liu et al., 2024) with a learning rate of $10^{-5}$, weight decay of 0.1, and no dropout. We use a batch size of 1024 examples, unless otherwise stated. We minimize the cross-entropy loss during training. Some graphs contain multiple paths from the start to the goal vertex. During training, we select one path uniformly at random as the ground truth when computing the loss. During evaluation, we consider the model's prediction to be correct if the output vertex lies on *any* path to the goal.

### 3.1.1 SENSITIVITY TO TRAINING DISTRIBUTION

We first investigate the effect of the training distribution on the transformer's ability to learn the search task. We do so by training one model on the naïve distribution, another model on the star distribution (where the number of spokes $k$ and spoke lengths $L$ are uniformly distributed), and a third model on the balanced distribution (where the lookahead $L$ is uniformly distributed), all with input size 128 tokens. Then, we evaluate the accuracy of each model on held-out test sets from the naïve, star, and balanced distributions, for all possible lookaheads $L$. We observe in Figure 2 that the model trained on the naïve distribution is not able to robustly handle graphs with larger lookaheads, showing low accuracy even for observed number of lookaheads (e.g., $L = 5$). This is due to the fact that the probability of generating graphs with larger lookaheads with the naïve distribution decreases exponentially, and so the model is not shown sufficient examples with large lookaheads. The model trained on the star distribution performs reasonably on examples from the balanced distribution but not as well on examples from the naïve distribution. In contrast, the model trained on the full balanced distribution performs near perfectly in all test settings, and generalizes to unobserved numbers of lookaheads. This result demonstrates that it is possible to teach a transformer to perform search almost perfectly on the space of lookaheads observed during training, when provided with an appropriate training distribution. Furthermore, the model trained on the balanced distribution with lookahead $\leq 12$ was able to generalize to lookaheads 13 and 14 but not on any larger lookaheads.

### 3.1.2 PROOF SEARCH IN NATURAL LANGUAGE

Our earlier experiments were conducted with a symbolic input representation for the graphs in each example. To demonstrate that our findings generalize to inputs expressed in natural language, we re-run our earlier experiment where we train small transformers from scratch, but with a modified input: Each edge (e.g., "vertex 1 connects to vertex 2") is expressed as a conditional sentence (e.g., "If Alex is a wumpus, then Alex is a vumpus") where each word and punctuation symbol is a token. The task is correspondingly modified: Given a start proposition (e.g., "Alex is a wumpus"), find the next step in the proof of a goal proposition. Thus, this modification defines a one-to-one mapping between graph search problems in the symbolic representation and reasoning problems in a natural

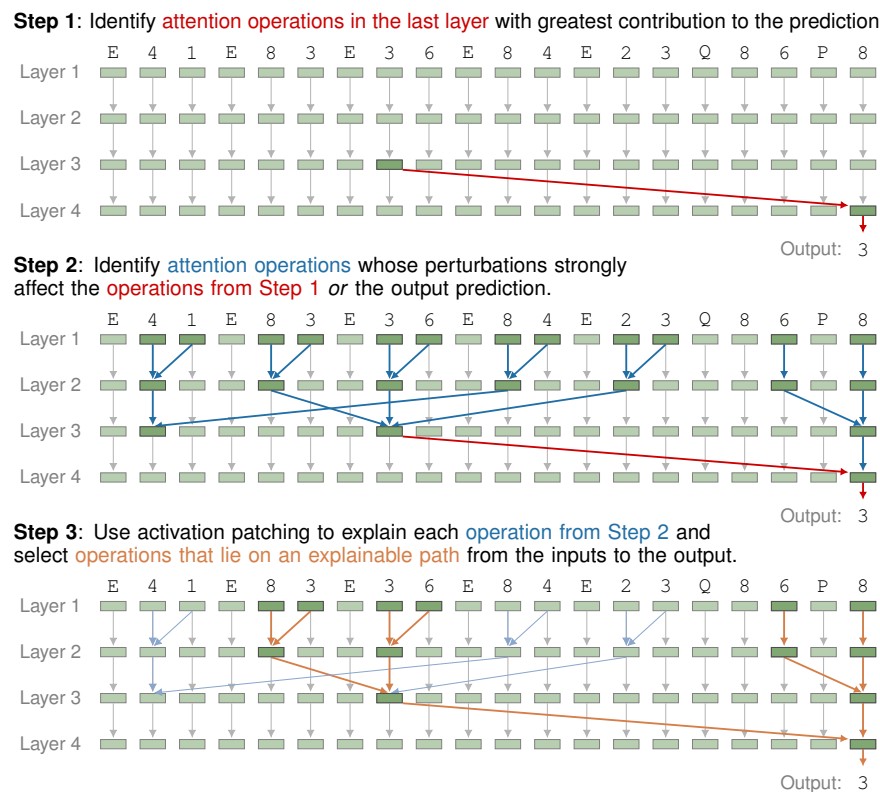

FIGURE 3: Overview of method to reconstruct the computation graph from a transformer for a specific input.

language representation (in implicational propositional logic). See Figure 1 for an example of this correspondence. We see in Figure 9 that the training behavior in this setting is qualitatively very similar to that in the graph search setting. The model has increasing difficulty learning the task as the graph size increases, especially in terms of FLOPs, as evident from the last row of the Figure.

## 4 MECHANISTIC INTERPRETATION OF TRANSFORMERS ON GRAPH SEARCH

We observed in the previous section that transformers are *sometimes* able to learn to search during training, and the resulting model is able to robustly and correctly answer almost any graph search problem in the input space. We aim to better understand the algorithm that the model acquired during training to solve the task, to determine whether and to what extent the model is utilizing a correct algorithm to solve the task, as opposed to a heuristic.

### 4.1 RECONSTRUCTING ALGORITHMS FROM INPUTS

In order to understand how the transformer learns to solve the graph search task, we develop a new method for mechanistically interpreting the model's behavior. Our method involves closely examining the model's behavior for a given input example in order to reconstruct a *computation graph* that explains the model's activations, attention patterns, and output prediction. Our method consists of the following steps, as depicted visually in Figure 3:

**I. Compute activations, attention weights, and output logits.** For a given input example, perform an ordinary forward pass to compute the activations, attention weights, and output logits.

**II. Identify important attention operations in the last layer.** For each element of the attention matrix in the last layer corresponding to the last token, perturb the weight by changing it to 0 and recomputing the logit of the model's original prediction. If the resulting decrease in the logit of the prediction is greater than a threshold parameter $\alpha$, then this attention operation is marked as *important*. Similarly perturb each weight by changing it to a large value[2] (the largest attention weight in the row of the attention matrix and renormalize). If the resulting decrease in the logit is greater than $\alpha$, then this operation is also marked as important.[3]

---

[2]We do this since we found that at some layers, a token will attend strongly to every other token *except* one token, and information is actually transferred from the exceptional token due to the small attention weight.

[3]Sometimes, this step yields no important operations. In such cases, as a fallback, the set of attention operations that cause the largest decrease in the logit of the model's prediction are marked as important.

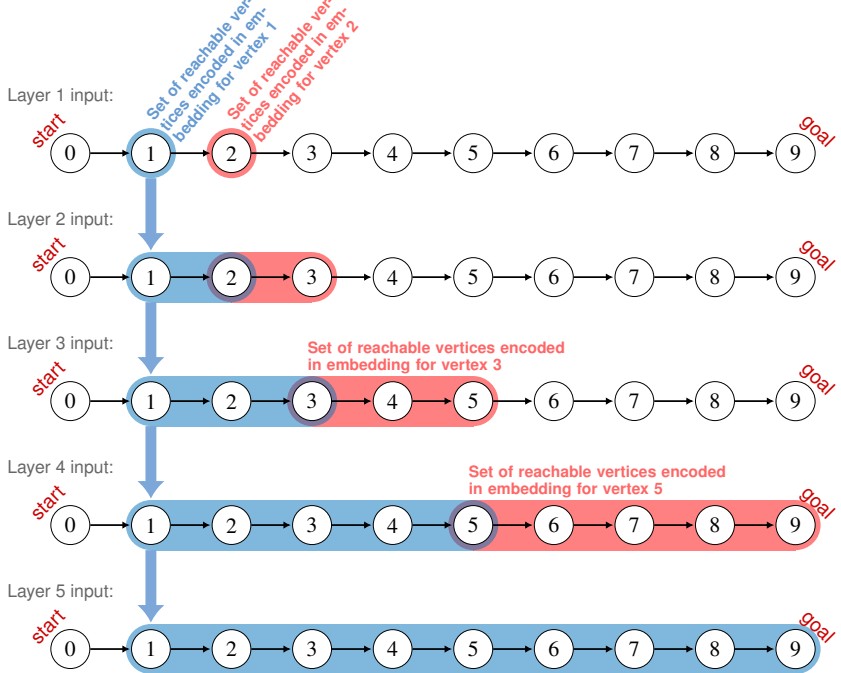

FIGURE 4: Visualization of the exponential path-merging algorithm, showcasing the layer-by-layer computation of the reachability of vertex 9 from vertex 1. We hypothesize that transformers learn this algorithm to search. In this algorithm, each token corresponding to a vertex stores information about which other vertices are reachable from this vertex (or from which vertices is this vertex reachable). For example, in layer 3, the model knows that vertex 3 is reachable from 1, and that 5 is reachable from 3, and computes that 5 is reachable from 1, as shown in the input to layer 4. We posit the model performs this computation for all vertices simultaneously.

III. **Identify important attention operations in all other layers.** For each element of the attention matrix in all layers *except* the last layer, perturb the weight by changing it to 0 or to a large value (i.e., setting the weight equal to the largest weight in that row and renormalizing). Perform a forward pass and inspect the log attention weight of each important operation in the last layer. If the resulting change in the log attention weight is greater than $\frac{\sqrt{d}}{\kappa_1}$, where $d$ is the model dimension and $\kappa_1$ is a sensitivity parameter, then we mark this attention operation as important. Similarly, if the resulting decrease in the logit of the output prediction is larger than $\alpha$, we mark this attention operation as important.

IV. **Explain each important attention operation.** For each important attention operation, let $j$ be the row and $i$ be the column of the corresponding entry in the attention matrix, and $l$ be the layer of the operation. We say token $i$ is the *source* token of this attention operation and token $j$ is the *target*. We use *activation patching* (Vig et al., 2020a; Zhang & Nanda, 2024) to determine which features of the input are causally significant for this operation. We test perturbations on two types of input features:

- **Token perturbations:** For each vertex ID $v$ of the input, we produce a perturbed input where $v'$ is substituted for $v$, where $v'$ is a vertex ID that does not appear in the original input.
- **Position perturbations:** For each position $i$ of the input, we produce a perturbed input where the positional embedding for the $i^{\text{th}}$ token is set to zero.

Suppose we perturb an input feature $f$. We then compute the forward pass on the perturbed input while freezing the attention matrices up to the layer of the current attention operation.[4] At layer $l$, we compute the dot products:

$$\tilde{Q}_j K_i^\top \quad \text{and} \quad Q_j \tilde{K}_i^\top, \tag{1}$$

where $\tilde{Q}_j$ is the perturbed query vector corresponding to token $j$, $Q_j$ is the unperturbed query vector, $\tilde{K}_i$ is the perturbed key vector corresponding to token $i$, and $K_i$ is the unperturbed key vector. We compare $\tilde{Q}_j K_i^\top$ to the original scaled dot product $Q_j K_i^\top$. If the resulting change in the dot product is greater than $\frac{\sqrt{d}}{\kappa_2}$ where $\kappa_2$ is a sensitivity parameter,[5] then we say that the

---

[4]We also freeze the ReLU activations in that any value that was set to zero by ReLU in the original forward pass will also be set to zero in the perturbed forward pass. The aim of freezing the previous layers is to measure the effect of the perturbation on the current layer *in isolation* of changes in behavior in preceding layers.

[5]We also require the change in dot product to be in the correct direction: If this attention operation has large attention weight, then we require the perturbed dot product to be smaller than the threshold, and vice versa for attention operations with small attention weight.

embedding of the token at index $j$ contains information about the perturbed feature $f$, and that this information is used by attention layer $l$ to perform this attention operation. Repeat this for all input features and we obtain the set of features $f_1^T, \ldots, f_v^T$ of the target embedding that causally affect this attention operation. Similarly, we compare $Q_j K_i^\top$ to determine whether information about the perturbed feature $f$ is encoded in the embedding of the token at index $i$ and is significant for this attention operation. Repeat this for all input features and we obtain the set of features $f_1^S, \ldots, f_u^S$ of the source embedding that causally affect this attention operation. The result of this step is a description of each important attention operation, showing *why* the operation happens. That is, what are the input features (token and positional embeddings) that are encoded in the source and target embeddings that causally affect the attention weight corresponding to this operation.

**V. Reconstruct the computation graph/circuit.** Starting from the first layer, let $t_k$ be the token at position $k$ of the input. We say each input vector "*explainably contains*" information about the token value $t_k$ and position $k$. Next, we consider the attention operations in the first layer. Suppose an attention operation copies source token $i$ into target token $j$, and depends on the source token embedding containing features $f_1^S, \ldots, f_u^S$ and depends on the target token embedding containing features $f_1^T, \ldots, f_v^T$ to perform this operation (as computed in Step **IV.**). We say this attention operation is *explainable* if the embedding of token $i$ explainably contains all features $f_1^S, \ldots, f_u^S$, and the embedding of token $j$ explainably contains all features $f_1^T, \ldots, f_v^T$. If the attention operation is explainable, we say the output embedding of the target token $j$ explainably contains the union of the features: $f_1^S, \ldots, f_u^S, f_1^T, \ldots, f_v^T$. We repeat this for every layer, computing all explainable attention operations throughout the model. Pseudocode for this procedure is shown in Algorithm 1. Finally, we filter out attention operations for which there does not exist a path of explainable attention operations to the output prediction (i.e., we can't explain how this operation is useful for the model's output on this example). The result is a computation tree, where each node corresponds to an embedding vector in some layer in the model, which explainably contains information about a set of input features, and where each edge corresponds to an explainable attention operation.

We apply the above method on a trained model repeatedly for different input examples. The result is a set of computation graphs/circuits, one for each input example, and we can perform further analysis on these circuits to describe the model's computation across many inputs. While this method is able to produce a fine-grained description of the processing in the transformer, it requires many forward passes[6]. Nonetheless, we are able to apply it to our smaller trained models.

## 4.2 EXPERIMENTS

We perform the above analysis on models trained on the balanced distribution that have achieved near-perfect test accuracy. We hypothesize that the transformer performs search on all vertices simultaneously, where the embedding for each vertex explainably contains information about the set of vertices reachable from the current vertex within a certain number of steps. At each layer, for each vertex, the attention mechanism copies from a source vertex that is at the edge of the current vertex's reachable set, computing the union of the reachable sets of both vertices and storing the resulting set in the embedding of the current vertex. Thus, the reachable set can theoretically double in size at every layer. In theory, the model may perform the search either in the forward or backwards direction: Rather than storing the set of reachable vertices, it may store the set of vertices from which the current vertex is reachable. A visual depiction of this algorithm is shown in Figure 4.

To test whether the transformer utilizes this algorithm, we perform the analysis described in Section 4.1 for multiple held-out inputs from both the naïve and balanced distributions (on a total of 2000 inputs; 100 for each lookahead). We set $\alpha = 0.4$, $\kappa_1 = 20$, and $\kappa_2 = 10$. For each input, we reconstruct and inspect the computation graph of attention operations. We categorize each attention operation into one of the following: (1) path-merge operations,[7] or (2) copy operations from vertices that are specifically reachable from either the start or the goal vertex. If the attention operation does not fall into either category, it is discarded. We say the input is *explained* by the path-merging algorithm if for every vertex along the path from the start to the goal vertex, there exists an unbroken sequence of path-merge operations that ultimately copy from the corresponding token in the first layer into the last token at the last layer.

---

[6]$Ln^2 mF$ where $L$ is the number of layers, $n$ is the model's input size, $m$ is the number of input examples, and $F$ is the number of perturbed features.

[7]We check for either "token-matching" or position-based path-merge operations. In token-matching, the attention head selects another token by looking for vertex IDs of overlapping sets of reachable vertices. In a position-based op, the attention head looks for vertices that are one step from a vertex in the reachable set.

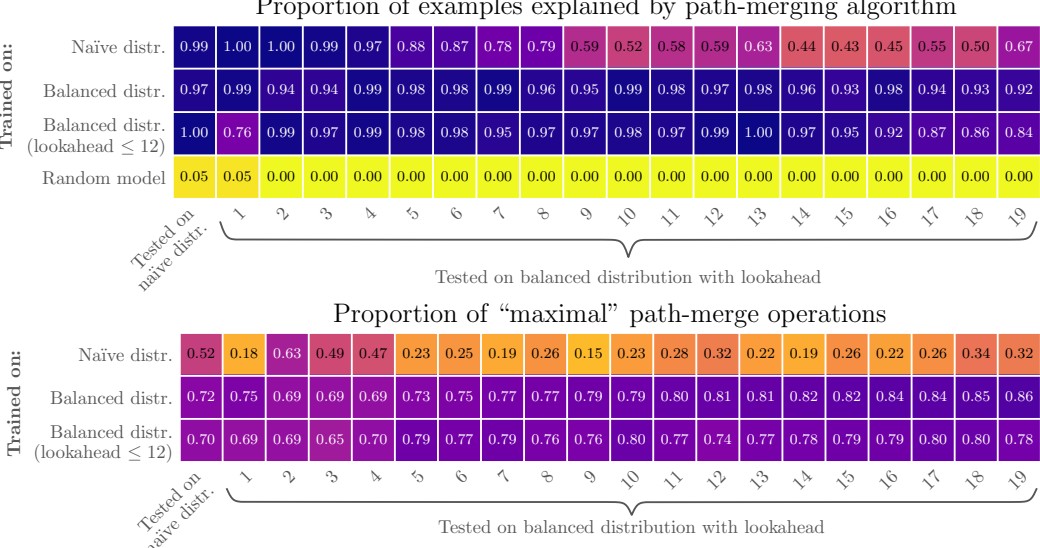

FIGURE 5: **(top)** The proportion of examples for which the path-merging algorithm was identified in the computation graph, as reconstructed using our mechanistic interpretability analysis. Each cell contains a random held-out sample of 100 examples. We perform our analysis on the same models as in Section 3.1.1 (and Figure 2). A randomly-initialized (untrained) model is shown in the last row as the control. **(bottom)** The proportion of path-merge operations that are "maximal," averaged over 100 random examples. We say a path-merge operation is maximal if it is merging the largest available reachable sets. This is in contrast with a suboptimal path-merge operation where one or both reachable sets are not the largest available at that layer.

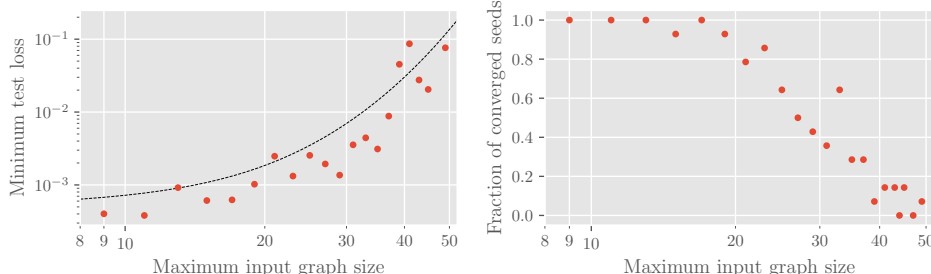

FIGURE 6: **(left)** The minimum test loss after training on 236M examples versus maximum input graph size (in vertices). For each maximum input graph size, we run 14 experiments with different random seeds. All models were trained on the balanced distribution. Test loss was evaluated on held-out examples from the naïve distribution. We only show results for models that have converged (i.e., it's training accuracy is greater than 0.995). **(right)** The fraction of models (initialized with different random seeds) that converged versus maximum input graph size after training on 236M examples.

The proportion of examples for which our method identifies the path-merging algorithm is shown in the top part of Figure 5. We observe that our method is highly specific, identifying the algorithm in the trained models but not in the random (untrained) model. Our analysis provides a more fine-grained view of the transformer's computation, and we are able to count individual path-merge operations and inspect whether they are "maximal" in the sense that they are merging the largest reachable sets that are available at that layer. For example, a path-merge operation that copies the embedding of vertex 2 into that of vertex 1 would be maximal if the embedding for 1 contains the reachable set $\{1, 2\}$ and the embedding for 2 contains the set $\{2, 3\}$, whereas the operation would be maximal if the embedding for 2 has the set $\{2\}$ (see the example in Layer 2 of Figure 4). We observe in the bottom portion of Figure 5 that the proportion of path-merge operations that are maximal is notably less than 1, indicating that the model does not learn maximal path-merge operations. And the model has no reason to do so since it is trained on lookaheads that do not align with a power of 2 and it has more layers than it needs to learn to search on the graphs with the largest lookaheads that can fit in its input. This explains why, in Figure 2, the model trained on lookaheads $L \leq 12$ is not able to fully generalize to larger lookaheads.

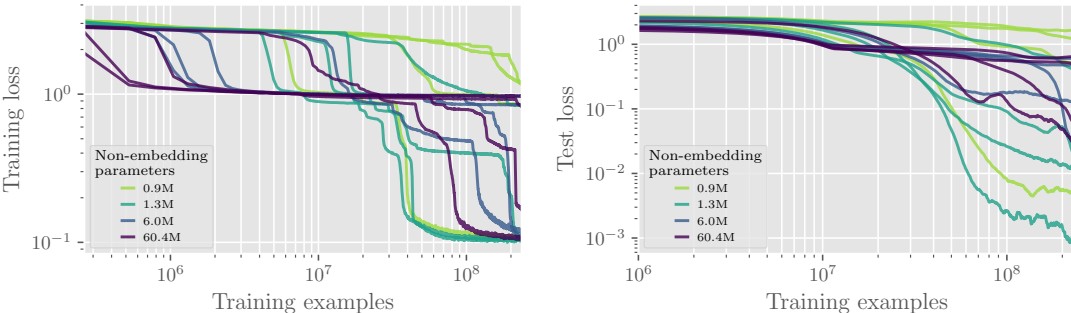

FIGURE 7: Training and test loss vs number of training examples seen, for models with varying numbers of non-embedding parameters. All models were trained on the balanced distribution. Test loss was evaluated on held-out examples from the naïve distribution. Test loss is smoothed by averaging over a window of 81 data points, where each data point is recorded every $2^{18} = 262\text{K}$ examples. 4 seeds are shown for each model size.

## 5 DOES SCALING HELP?

Even if it is possible to train a transformer to search, it is unclear how this ability scales with respect to input graph size and model scale. We investigate scaling behavior by running two sets of experiments on small transformer models: (1) training models on increasing graph sizes, and (2) training models with increasing model dimension $d$. We observe large variance in performance across different initial random seeds, which has been observed in other tasks (Kim & Linzen, 2020; Zhou et al., 2024). Thus, in each of these experiments, we train the models using multiple seeds on examples from the balanced distribution, and measure the minimum loss on a held-out test set generated by the naïve distribution. We set the batch size to 256 examples due to GPU memory limitations.

In Figure 6, we observe that, when the number of layers is fixed to 8 and the hidden size 16, as the maximum input graph size is increased, the likelihood that the model learns the training distribution (i.e., reaches accuracy $\geq 0.995$) becomes vanishingly small. In addition, as the maximum input graph size increases, the minimum test loss over 14 seeds grows at an increasing rate. See Figure 10 in the Appendix for a more detailed visualization of the training dynamics versus input graph size.

To determine whether larger models can more easily learn to search on large input graphs, we fix the input graph size to 31 and train models of widely varying sizes. In Figure 7, we see that while larger models are able to more quickly find the local minimum (at loss near $10^0$), there is no discernible pattern between the size of the model and the amount of training needed to find the global minimum.

We also experiment with decoder-only models with learned token embeddings and *rotary positional embeddings* (Su et al., 2024), which are multiplied rather than concatenated, and are more predominant in contemporary LLMs. But we find that this does not change the model's scaling behavior (see Section A.5).

## 6 DOES IN-CONTEXT EXPLORATION (I.E., CHAIN-OF-THOUGHT) HELP?

Though we have shown that transformers are not able to learn to perform search for larger input graphs, they may be able to learn to search if permitted to take intermediate steps, akin to chain-of-thought prompting. To test this, we repeat our earlier experiments with two "prompting" approaches: (1) depth-first search, and (2) selection-inference.

### 6.1 DEPTH-FIRST SEARCH

We generate graphs and perform a depth-first search (DFS) from a randomly-selected vertex to a random goal vertex. From the sequence of visited vertices (i.e., *DFS trace*), we randomly select a "current" vertex. Each input to the model is: (1) the graph, as a list of edges, and (2) the sequence of visited vertices up to and including the current vertex. The model's task is to predict the next vertex in the DFS trace.[8]

In our earlier search task, the edges of the graph always appeared in the same token position across inputs, since the current and start vertices were identical. However, in this modified task, the se-

---

[8]Again note that there may be multiple correct predictions, since there are typically many correct DFS traces for a given graph. Similar to the setup in Section 3.1, we randomly select one of them to be the ground truth label when computing the cross-entropy loss during training. But during evaluation, we allow the model to make any prediction that follows a valid DFS trace.

quence of visited vertices can vary in length. In order to avoid complicating the task for the model, we add padding to the input, between the graph and the sequence of visited vertices: The right-most tokens are reserved for the sequence of visited vertices. All other tokens contain the graph's edges, just as in the original search task. See the example in Figure 13. The graphs in the training distribution are sampled such that the backtrack distance is uniformly distributed, analogous to the balanced distribution in our earlier experiments (see Section A.7 for details on the graph distribution).

In the experiments, we first fix the model size, setting the number of layers to 3, and vary the input graph size. As evident in the top row of Figure 14, the model is able to learn the training distribution across all tested graph sizes, suggesting that only a constant number of layers is needed to learn the DFS task. However, the model struggles as the graph size increases. Next, we fix the maximum graph size to 35 vertices and instead vary the model size. In the middle and bottom rows of Figure 14, we observe that increasing model scale does not help the transformer to learn this task more easily. Interestingly, we also find that larger models are able to learn the task from *fewer* training examples. However, the benefit from scale disappears when considering the cost of training larger models: Larger models require many more FLOPs to learn the task than the smaller models.

## 6.2 SELECTION-INFERENCE

In this setting, each search step is decomposed into two subtasks: (1) given a graph and a list of visited vertices, **select** a visited vertex that has an unvisited child vertex, and (2) given a selected vertex, predict (i.e., **infer**) an unvisited child vertex. Starting from the start vertex, if these two subtasks are repeated sufficiently many times, the goal vertex will be found. To construct a graph distribution that is analogous to the balanced distribution, we define two variables: (1) the *frontier size* $F$, which is the number of visited vertices that have unvisited children, and (2) the *branch count* $B$, which is the number of child vertices of the current vertex (in an inference step). We generate graphs such that the pair $(F, B)$ is uniformly distributed. Each input consists of: (1) the graph, and (2) the sequence of visited edges.[9] See Section A.10 for further details.

In the experiments, we first fix the model size, setting the number of layers to 4, and vary the input graph size. We again note from the top row of Figure 16 that the model struggles as the graph size increases. Next, we fix the maximum graph size to 45 vertices and vary the model size. We note in Figure 17 that increasing model scale does not help to learn the task. Therefore, transformers struggle to learn to perform DFS search and selection-inference on larger graphs and additional scaling does not seem to make it easier.

## 7 CONCLUSION

Through the use of graph connectivity as a testbed, we found that transformers can learn to search when given the right training distribution. We developed and applied a new mechanistic interpretability technique on the trained model to determine the algorithm that the model learned to perform search. We found that the model uses an exponential path-merging algorithm, where the embedding of each vertex stores information about the set of reachable vertices from that vertex. As the input graph size increases, the transformer has ever-increasing difficulty in learning the task, and increasing the scale of the model does not alleviate this difficulty. Lastly, even if the model is permitted to use intermediate steps, they still struggle on larger graphs, regardless of scale.

It is possible that scaling to *much* larger model sizes may lead to emergent searching ability. Alternate training procedures may help transformers to more easily learn to search, such as curriculum learning. Alternate architectures may help as well, such as looped transformers. While the path-merging algorithm is able to explain almost all examples for the trained models, there may be other algorithms or heuristics that the model simultaneously utilizes on some examples. Our mechanistic analysis has potential broader applications in reasoning: Some form of the path-merging algorithm may be used by transformers in more general reasoning tasks. In such a case, the representation of each fact would store information about the set of facts *provable* from the current fact. Our mechanistic interpretability tools may be useful in other settings, as well, where they may help to uncover the algorithms that transformers learn to solve other tasks. Though additional work is welcome to improve the scalability of our analysis to larger models, our analysis can provide insights on smaller models that can be tested separately in larger models.

---

[9]Similar to the DFS task, we add padding to ensure the graph edges appear in the same positions across examples.

ACKNOWLEDGMENTS

We thank Emmanouil Antonios Platanios, Jannik Brinkmann, Dan Friedman, Arvid Frydenlund, Gonzalo Gonzalez-Pumariega, and Will Merrill for their invaluable insight and discussion. This work was supported by Open Philanthropy, and in part through the NYU IT High Performance Computing resources, services, and staff expertise, and the Rosen Center for Advanced Computing at Purdue University.

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

# A APPENDIX

## A.1 GRAPH GENERATION DETAILS

### A.1.1 NAÏVE DISTRIBUTION

To sample a graph from this distribution, we first sample the number of vertices

$$|V| \sim \text{Uniform}(\{3, \ldots, V_{\max}\}), \tag{2}$$

where $V_{\max}$ is the maximum number of vertices that can fit the input. Then, for each $i = 1, \ldots, |V|$, we sample a number of parent vertices $n_i^{\text{parents}}$ from $V_1, \ldots, V_{i-1}$:

$$n_i^{\text{parents}} = \begin{cases} 1 & \text{with probability } \frac{5}{8}, \\ 2, 3, \text{ or } 4 & \text{with probability } \frac{1}{8}. \end{cases} \tag{3}$$

Note that this differs from Erdős–Rényi where the number of parents is geometrically distributed. We want to avoid generating overly-dense graphs where the lookahead is too small, and so we choose to sample fewer parents per vertex.

Finally, we sample the parents of $V_i$ uniformly without replacement from $\{V_1, \ldots, V_{i-1}\}$ until we have sampled $n_i^{\text{parents}}$ vertices, or we have sampled all available vertices. We draw an edge from each sampled vertex to $V_i$.

After generating the graph, we randomly permute the vertex IDs so that the IDs contain no information about the graph topology. Observe that this distribution can generate any directed graph with maximum in-degree 4 in topologically-sorted order, and therefore, it can generate any DAG with maximum in-degree 4.

We select the start and goal vertices uniformly at random from $V$. If there is no path from the start to the goal vertex, or if the example does not fit within the model input, we reject the sample and try again.

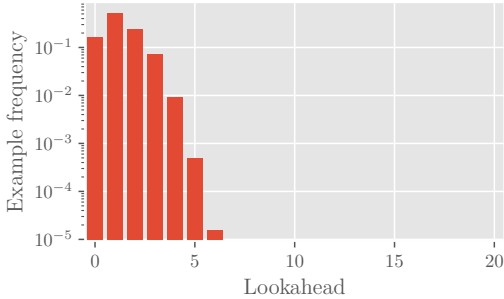

FIGURE 8: Histogram of lookaheads of 10M graphs sampled from the naïve distribution (note the log y-axis). The number of vertices $n$ is 41. Lookaheads $> 6$ are possible, but astronomically unlikely.

### A.1.2 BALANCED DISTRIBUTION

Each graph is sampled according to the following procedure: Given a lookahead parameter $L$, we start by creating a linear chain of vertices containing $L$ edges. The first vertex in this chain is set as the start vertex, and the last vertex is the goal vertex. We then sample a number of additional "branches":

$$B \sim \text{Uniform}(\{1, \ldots, \lfloor \tfrac{V_{\max} - 1}{L} \rfloor\}). \tag{4}$$

Next, we sample the total number of vertices:

$$|V| = \min\left\{ L(B+1) + 1 + u, V_{\max} \right\}, \tag{5}$$

where $u \sim \text{Uniform}(\{0, \ldots, 6\})$. We create $B$ additional chains of vertices: For $i = 1, \ldots, B$, we create a chain of vertices where the length of the chain in edges $l_i$ is given by

$$l_i \sim \begin{cases} L, & \text{if no additional vertices are available, i.e., } \sum_{j=1}^{i-1} l_j - L(B - i - 1) + 1 = |V|, \\ \text{Uniform}(\{L, L+1\}), & \text{otherwise.} \end{cases} \tag{6}$$

Each additional branch is added to the start vertex.

While the graphs resulting from the above process will require the model to search at least $L$ steps to find the goal, the graphs still admit heuristics since the start vertex is always the singular source vertex of the graph (i.e., has zero in-degree), and all other vertices have in-degree exactly equal to 1. To prevent such heuristics, we sample additional vertices as follows: We create an "incoming" linear chain with length $l^{\text{in}} \sim \text{Uniform}\{0, \ldots, |V| - \sum_{j=1}^{B} l_j + 1\}$. In contrast with the other branches, the start vertex is located at the *end* of this chain. Finally, to increase the degrees of the vertices in the graph, we create additional vertices until we have a total of $|V|$ vertices. For each new vertex $V_i$, sample a number of child and parent vertices:

$$n_i^{\text{children}} \sim \text{Uniform}\{0, 1, 2, 3\}, \tag{7}$$

$$n_i^{\text{parents}} \sim \text{Uniform}\{\mathbb{1}\{n_i^{\text{children}} = 0\}, \ldots, 3\}, \tag{8}$$

where $\mathbb{1}\{x\}$ is the indicator function whose value is 1 if $x$ is true, and 0 otherwise. We sample $n_i^{\text{children}}$ child vertices from $V_1, \ldots, V_{i-1}$ without replacement where the probability of sampling $V_j$ is proportional to $\frac{1}{2} + \deg^-(V_j)$ where $\deg^-(v)$ is the in-degree of $v$ (at the time of sampling). Similarly, we sample $n_i^{\text{parents}}$ parent vertices from $\{V_1, \ldots, V_{i-1}\} \setminus \text{descendants}(V_i)$ without replacement where the probability of sampling $V_j$ is proportional to $\frac{1}{2} + \deg^+(V_j)$ where $\deg^+(v)$ is the out-degree of $v$. Note we avoid sampling from the descendants of $V_i$ in order to avoid creating cycles. We chose this sampling scheme in order to produce a handful of vertices with high in- or out-degree, and to prevent the model from exploiting a heuristic when all vertices have low degree. This distribution is an example of a scale-free distribution over directed graphs (Bollobás et al., 2003).

As in the naïve distribution, after generating the graph, we randomly permute the vertex IDs so that the IDs contain no information about the graph topology. If the resulting graph does not fit within the model input, we reject the sample and try again.

## A.2 NATURAL LANGUAGE PROOF SEARCH RESULTS

Figure 9 shows the results of our experiments on the natural language proof search task, as described in Section 3.1.2.

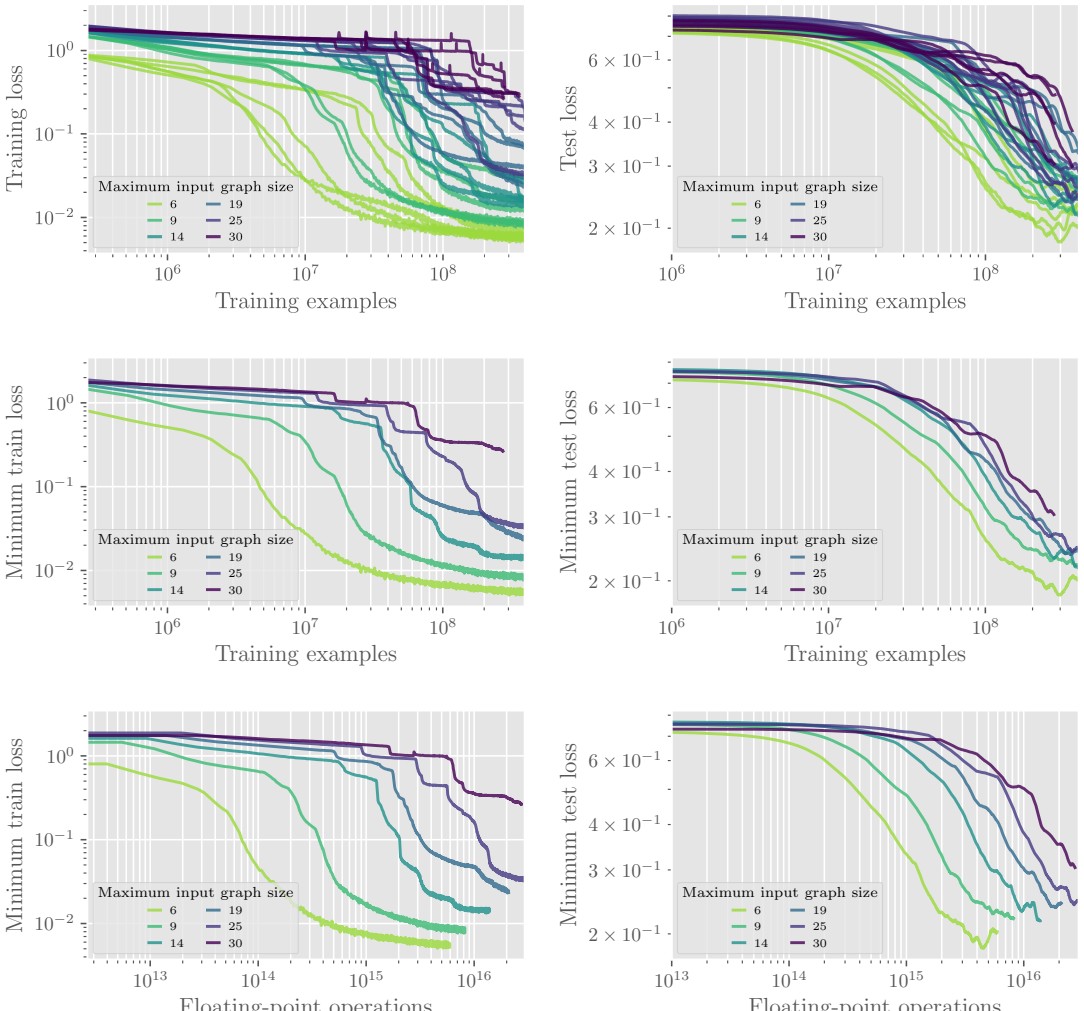

FIGURE 9: Training and test loss vs number of training examples seen, for models trained on the natural language proof search task, with varying maximum input graph sizes. All models were trained on the balanced distribution. Test loss was evaluated on held-out examples from the naïve distribution. Test loss is smoothed by averaging over a window of 81 data points, where each data point is recorded at every $2^{18} = 262\text{K}$ examples. In the top row, 5 seeds are shown for each maximum input graph size. In the middle and bottom row, the minimum loss over the seeds is shown. In the bottom row, the x-axis is rescaled as FLOPs.

## A.3 ALGORITHM RECONSTRUCTION PSEUDOCODE

---

**Algorithm 1:** Pseudocode of the procedure to compute the set of explainable attention operations of a transformer $\mathcal{M}$ for a given input $x$.

---

```
 1  function reconstruct_computation_graph(input example x, transformer M)
 2      let L be the number of layers in M
 3      initialize N_{l,i} as an empty set for all l ∈ {0,...,L} and all i ∈ {1,...,|x|}
 4      initialize E as an empty set
 5      for i ∈ 1,...,|x| do
 6          add token feature x_i to N_{0,i}
 7          add position feature i to N_{0,i}
 8      for l ∈ 1,...,L do
 9          for i ∈ 1,...,|x| do
10              for j ∈ 1,...,|x| do
11                  let e represent the attention operation at layer l that copies from source token i to target token j
                        /* this is computed in Step IV. as described in Section 4.1 */
12                  let f_1^S,...,f_u^S be the features in token i on which e depends
13                  let f_1^T,...,f_v^T be the features in token j on which e depends
14                  if {f_1^S,...,f_u^S} ⊆ N_{l-1,i} and {f_1^T,...,f_v^T} ⊆ N_{l-1,j}
                        /* mark this attention operation as explainable */
15                      add e to E
16                      add {f_1^S,...,f_u^S} ∪ {f_1^T,...,f_v^T} to N_{l,j}

            /* N_{l,i} now contains the set of features that are explainably
               contained in the embedding vector of token i at layer l */
            /* E contains the set of all explainable attention operations */
            /* next, filter the explainable attention operations that are not
               useful for the model's prediction */
17      let S be an empty stack
18      push (L, n) onto S
19      initialize E* as an empty set
20      while S is not empty do
21          pop (l, j) from S
22          for attention operation e ∈ E at layer l whose target is token j do
23              add e to E*
24              let i be the source token of the attention operation e
25              push (l − 1, i) onto S
26      return N, E*
```

---

## A.4 ADDITIONAL SCALING RESULTS

Figure 10 provides a visualization of the training dynamics of the transformer when trained with varying maximum input graph sizes. Figure 6 focuses on the slice at 236M training examples.

## A.5 SCALING DECODER-ONLY MODELS WITH ROTARY POSITIONAL EMBEDDINGS

We repeat the experiments in Section 5 on decoder-only transformers with learned token embeddings (initialized randomly from a Gaussian distribution) and rotary positional embeddings (RoPE). In our other experiments, the token embedding was concatenated with the positional embedding. In contrast, RoPE positional embeddings are multiplied with the token embedding in these experiments. For each maximum input graph size, we set $d_{\text{model}}$ to be the same as the corresponding experiment in Section 5 and A.4 for the same maximum input graph size. We observe in Figure 11 that decoder-only models with RoPE similarly struggle to learn to search on larger graphs. In addition, in Figure 12, we see that increasing the model scale does not help the model to learn the task more easily.

## A.6 DFS EXAMPLE

Figure 13 shows a DFS example, as described for the task in Section 6.1.

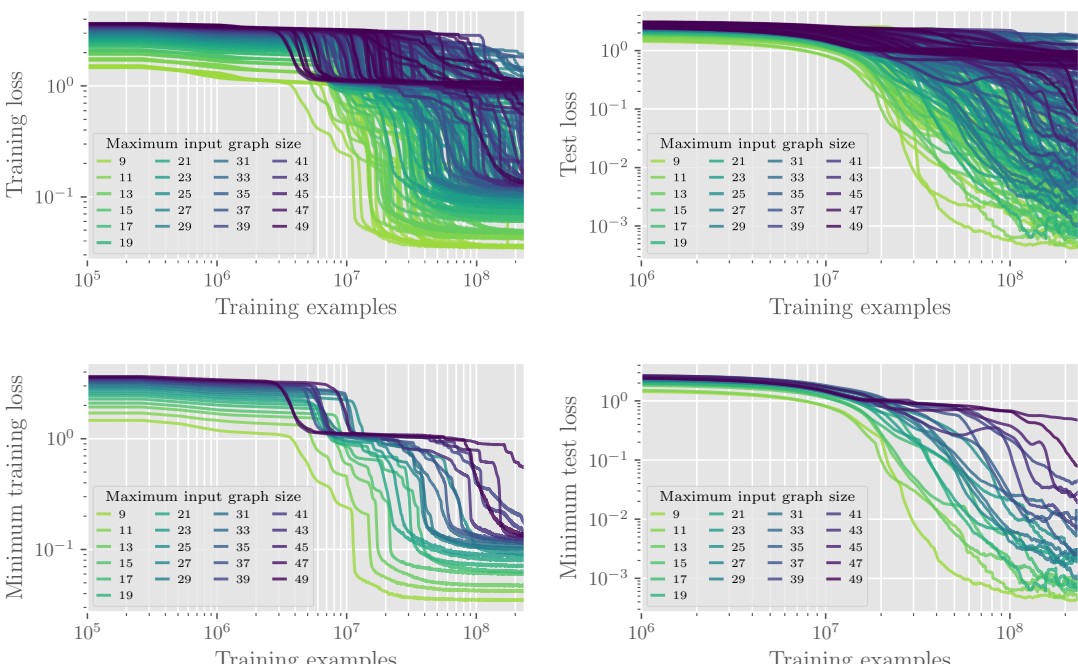

FIGURE 10: Training and test loss vs number of training examples seen, for models with trained on varying maximum input graph sizes. All models were trained on the balanced distribution. Test loss was evaluated on held-out examples from the naïve distribution. Test loss is smoothed by averaging over a window of 81 data points, where each data point is recorded at every $2^{18} = 262K$ examples. In the top row, 14 seeds are shown for each maximum input graph size. In the bottom row, the minimum loss over the seeds is shown.

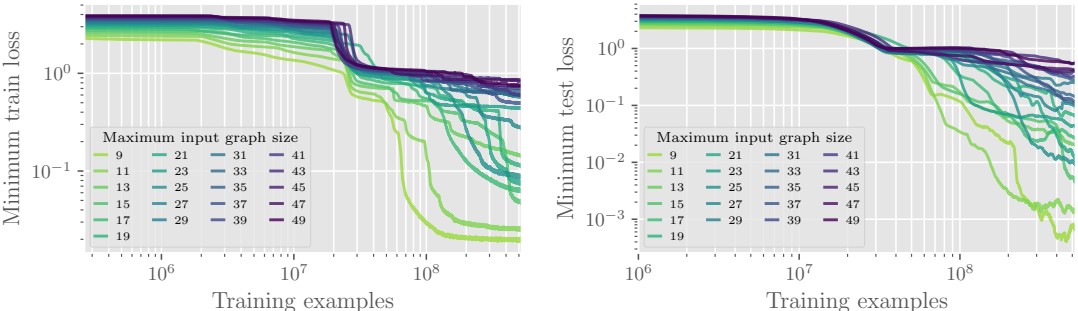

FIGURE 11: Training and test loss vs number of training examples seen, for decoder-only transformers using rotary positional embeddings. Test loss was evaluated on held-out examples from the naïve distribution. We fix the model size and vary the maximum input graph size. All models were trained on the balanced distribution. Test loss is smoothed by averaging over a window of 81 data points, where each data point is recorded at every $2^{18} = 262K$ examples. For each point, we plot the minimum loss over 2 seeds.

### A.7  GENERATING DFS EXAMPLES WITH SPECIFIC BACKTRACK DISTANCES

To generate graphs with a specific backtrack distance $B$, we first sample a DAG from the naïve distribution. We then divide the (topologically-sorted) graph into two subgraphs: the first $|V| - B$ vertices form the first subgraph $G_1$, and the last $B$ vertices form the second subgraph $G_2$. $G_1$ will contain the goal vertex, and $G_2$ will contain the list of vertices visited so far (i.e., the DFS trace).

Let $G_{1,-1}$ be the last vertex in $G_1$ (in the topological ordering) and let $G_{2,1}$ be the first vertex in $G_2$. Next, we select the start vertex $s$: If $G_{1,-1}$ is the only parent vertex of $G_{2,1}$, we sample the start vertex uniformly at random from $G_1 \setminus \{G_{1,-1}\}$. If not, then we sample a start vertex uniformly at random from parents($G_{2,1}$) $\setminus \{G_{1,-1}\}$ (since we need to leave at least one vertex in $G_1$ to be the goal). Then we sample the goal vertex $g$ uniformly at random from the set of vertices in $G_1$ that come after $s$.

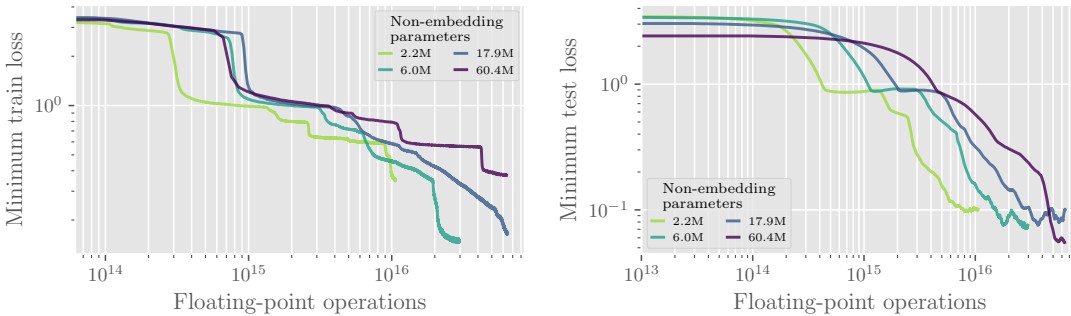

FIGURE 12: Training and test loss vs FLOPs, for decoder-only transformers using rotary positional embeddings. Test loss is computed on held-out examples from the naïve distribution. We fix the maximum input graph size to 31 vertices and vary the model size. All models were trained on the balanced distribution. Test loss is smoothed by averaging over a window of 81 data points, where each data point is recorded at every $2^{18} = 262K$ examples. For each point, we plot the minimum loss over 2 seeds.

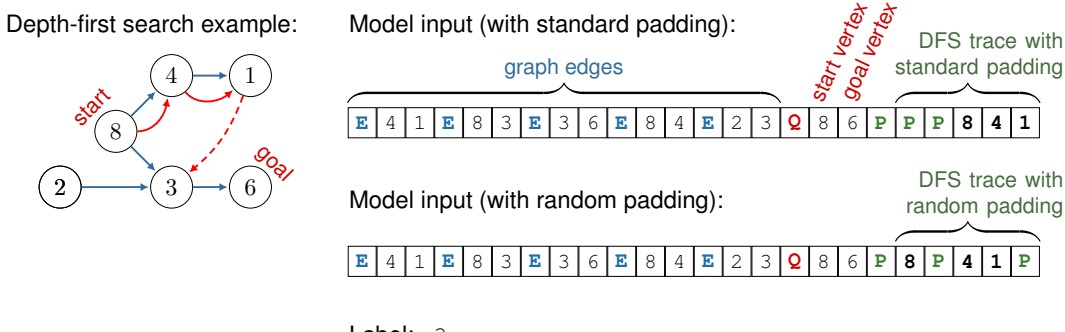

FIGURE 13: **(left)** Example of a depth-first search example on a directed acyclic graph where the model has visited the vertices 8, 4, and 1 so far. **(right)** The corresponding transformer input and output label. We experiment with two padding methods: (1) standard padding where the DFS trace is left-padded, and (2) random padding where padding is randomly inserted between vertices in the trace.

We have to make sure there exists a path from $g$ to every vertex in $G$ that comes after $s$. Iterating over the vertices in $G$ from left to right, starting with the vertex right after $g$, if there is no path from $s$ to that vertex, we add an edge between $s$ and that vertex.

Now that we have generated the graph, we next produce the DFS trace: We start with $s$ and visit every vertex in $G_2$. The next correct step in the DFS algorithm would be to backtrack from a vertex in $G_2$ to any child vertex of $s$ that is in $G_1$. Thus, the backtrack distance of this example is $|G_2| = B$.

As with the other graph and DFS example distributions, we randomly permute the vertex IDs as the last step.

## A.8 DFS SCALING

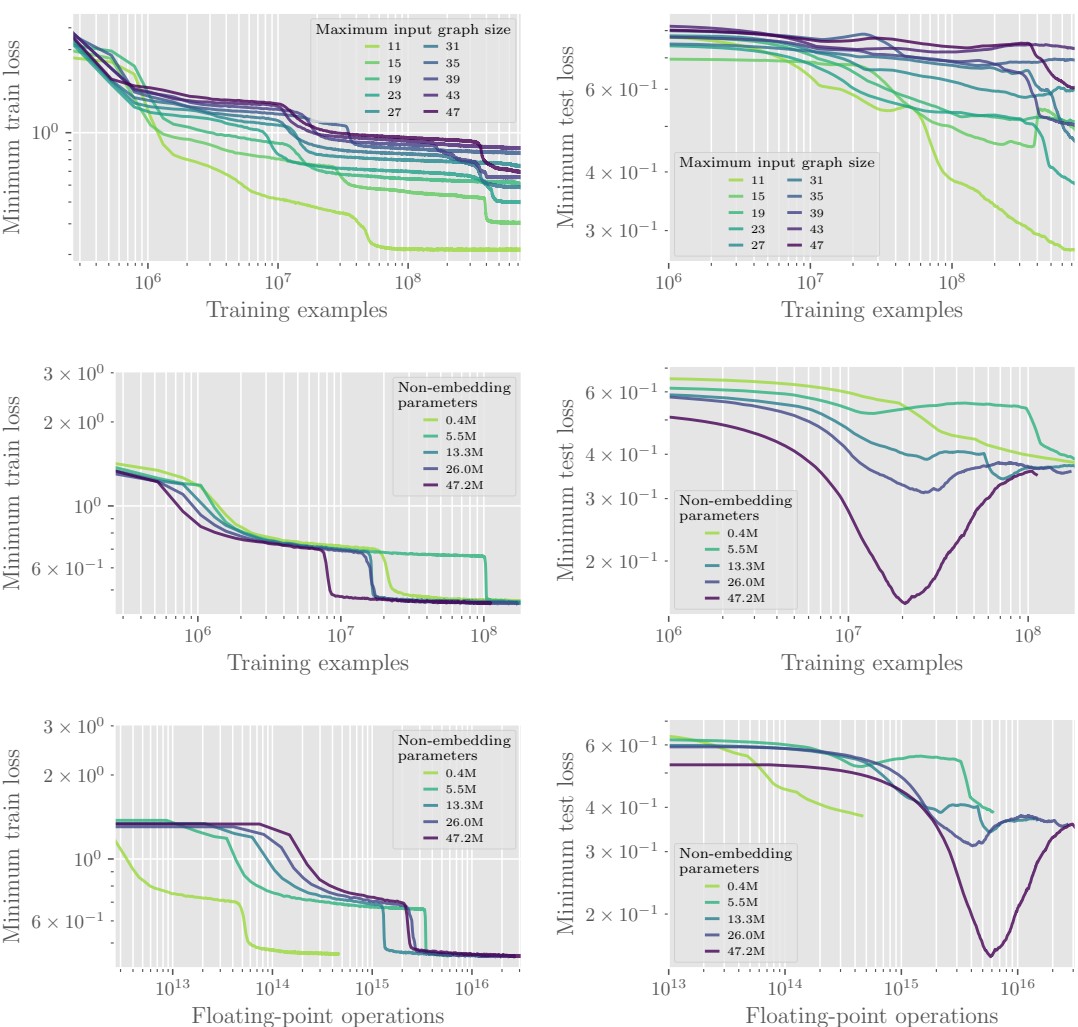

FIGURE 14: Training and test loss vs number of training examples seen, for models trained on the depth-first search task. Test loss is computed on held-out examples from the balanced distribution with backtrack 8. In the top row, we fix the model size and vary the maximum input graph size. In the middle and bottom rows, we fix the maximum input graph size and vary the model size. Note the x-axis in the bottom row is FLOPs. All models were trained on the balanced distribution. Test loss is smoothed by averaging over a window of 81 data points, where each data point is recorded at every $2^{18} = 262K$ examples. For each point, we plot the minimum loss over 15 seeds.

## A.9 NON-STANDARD PADDING IN DFS

We train a 7-layer transformer with input size 128 on the DFS task and evaluate on held-out examples with various backtrack distances. The results are shown in the top row of Figure 15. We observe that

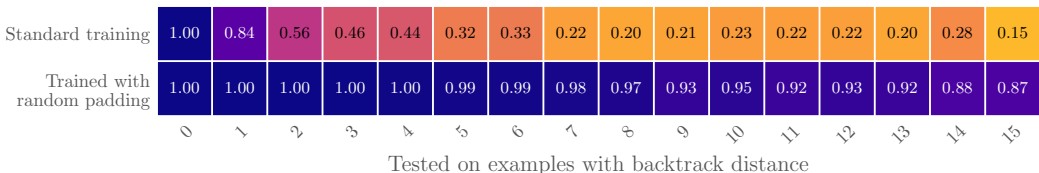

FIGURE 15: Accuracy of model trained on the depth-first search task with and without random padding. All evaluation is performed on held-out examples. Both models were trained on 746M examples.

the backtrack distance in the DFS search task is analogous to the lookahead distance in the original search task, where the probability of generating examples from the naïve distribution with higher backtrack distances is vanishingly low. However, unlike the original search task, we can augment training examples by adding padding between vertices in the sequence of visited vertices to help teach the model to backtrack greater distances. More precisely, starting with the right-most token, we add $k$ padding tokens where $k \sim \text{Uniform}(0, \ldots, T_{\text{available}})$ where $T_{\text{available}}$ is the total number of available padding tokens. We repeat this process for every token, from right to left. We observe in the second row of Figure 15 that when trained on these randomly-padded inputs, the model more robustly learns to backtrack to greater distances. Unlike in the original search task where we carefully developed the balanced distribution to train the model to search to greater lookaheads, a simple post-hoc augmentation of the examples was sufficient to successfully train the model to perform DFS search.

We note that while random padding may help the model to generalize to larger backtrack distances than those shown in training, they do not help the model to learn to search on larger graphs, as our scaling experiments have shown. In the scaling experiments, the training distribution contains examples of all possible backtrack distances, uniformly distributed.

### A.10 GENERATING SELECTION-INFERENCE EXAMPLES

To generate a selection-inference example with a given frontier size $F$ and branch count $B$, we first sample the graph size $|V|$ (number of vertices) from $\text{Uniform}(\{2, \ldots, E_{\text{max}} - F + 1\})$, where $E_{\text{max}}$ is the maximum number of edges that fit for the given transformer input size. Next, we arrange $|V|$ vertices from left to right (in topological order) without edges: $V_1, \ldots, V_{|V|}$. We select the index of the "current" vertex from $c \sim \text{Uniform}(\{1, \ldots, |V| - B\})$. We then sample the indices of the start and goal vertices:

$$s \sim \text{Uniform}(\{1, \ldots, \min(c, |V| - F)\}), \tag{9}$$

$$g \sim \text{Uniform}(\{\max(c + 1, s + F), \ldots, |V|\}). \tag{10}$$

**Add initial edges:** Next, we iterate over each vertex in the graph $V_i$, from left to right, and add edges as follows: First sample a number of parent vertices for $V_i$ from $n_i^{\text{parents}} \sim \text{Uniform}(\{0, \ldots, \lfloor \min(i - 1, \frac{n}{24}) + 1 \rfloor\})$ where $n$ is the transformer input size (in tokens). Next, we sample parent vertices from among $\{V_1, \ldots, V_{i-1}\}$ one at a time, with probability proportional to the out-degree of each vertex. We add an edge between the selected parent and $V_i$ and repeat until we have $n_i^{\text{parents}}$ parent vertices. However, if one of the potential parents is the current vertex $V_c$, and the number of child vertices of $V_c$ is $B$, we exclude it from the set of potential parents, as we want to ensure the branch count of $V_c$ is not larger than $B$.

**Construct the frontier:** Next, we sample the *frontier* vertices, which will be the vertices that have unvisited child vertices. $V_s$ and $V_c$ are automatically added to the set of frontier vertices. We sample the remaining vertices from $\{V_{s+1}, \ldots, V_{|V|}\} \setminus \{V_g\}$ uniformly at random until we have $F$ frontier vertices. We then perform selection-inference from $V_s$, selecting edges to explore uniformly at random, but we avoid selecting an outgoing edge from any frontier vertex, and we perform the search until no available edges remain. Let $\mathcal{E}$ be the list of visited edges (in the order they were visited). Note that there may still exist frontier vertices that have not been reached in $\mathcal{E}$. For each of these frontier vertices $V_i$: We randomly select an ancestor $V_a$ that has been reached in $\mathcal{E}$ and replace a random parent of $V_i$ with $V_a$, and add the edge $V_a \to V_i$ into $\mathcal{E}$. However, it is possible that there is no path from $V_s$ to $V_i$, in which case no ancestor of $V_i$ has been reached in $\mathcal{E}$. In this case, we select a vertex from $\{V_s, \ldots, V_{i-1}\}$ that has been reached in $\mathcal{E}$, uniformly at random, and add it as a parent of $V_i$. The new edge is added to $\mathcal{E}$. Note that each time an edge is added to $\mathcal{E}$, we move it into a random valid position.

**Ensure each frontier vertex has an unvisited child:** At this point, we have guaranteed that every frontier vertex has been reached in $\mathcal{E}$. Next, we have to ensure that every frontier vertex has at least one unvisited child vertex. For each frontier vertex $V_i$ without unvisited child vertices, we select a new child vertex $V_j$ from $\{V_{i+1}, \ldots, V_{|V|}\}$ that has not been reached in $\mathcal{E}$. Next, we randomly select a parent of $V_j$ that has not been reached in $\mathcal{E}$ and replace it with $V_i$. If $V_j$ has no such parent, we simply add the edge $V_i \to V_j$.

**Make sure the current node has $B$ child vertices:** Next, we want to ensure that $V_c$ has exactly $B$ child vertices. First, we add frontier vertices as children of $V_c$: we select a frontier vertex $V_f$ from $\{V_{c+1}, \ldots, V_{|V|}\}$ uniformly at random. If this vertex does not already have an edge from $V_c$, we add one. We replace the edge in $\mathcal{E}$ containing $V_f$ as the target with the new edge $V_c \to V_f$. We repeat

until $\max(0, 2F + B - E_{\max})$ frontier vertices are children of $V_c$ (or there are no further frontier vertices available). Then, we add non-frontier vertices as children of $V_c$: select a non-frontier vertex $V_j$ from $\{V_{c+1}, \ldots, V_{|V|}\}$ uniformly at random. We randomly sample a parent of $V_j$ that has not been reached in $\mathcal{E}$ and replace it with $V_c$. If $V_j$ has no such parent, we simply add the edge $V_c \to V_j$. Repeat until $V_c$ has $B$ child vertices.

**Make sure the goal vertex is reachable from the start vertex:** If $V_g$ is not reachable from $V_s$, select a random reachable vertex $V_i$ such that $i < g$ and add the edge $V_i \to V_g$.

**Remove some superfluous edges:** We next remove a number of superfluous edges (i.e., edges that are not in $\mathcal{E}$, are not needed to keep $V_s$ and $V_g$ connected, or are not needed to connect frontier vertices to an unvisited child vertex). If there are more than $E_{\max}$ edges, we remove superfluous edges randomly until $E_{\max}$ edges remain. Otherwise, we randomly remove $n$ superfluous edges where $n$ is the number of edges we have added since adding the initial edges.

**Add more edges to $\mathcal{E}$:** Next, we continue the selection-inference procedure from earlier to add additional edges to $\mathcal{E}$, taking care that each frontier vertex still has at least one unvisited child vertex. We continue until we have $n^{\mathcal{E}}$ edges, where $n^{\mathcal{E}} = i$ with probability proportional to $i$ and $n^{\mathcal{E}} \in \{|\mathcal{E}|, \ldots, E_{\max}\}$. This step helps to make sure the size of $\mathcal{E}$ is more uniformly distributed.

As with the other graph and DFS example distributions, we randomly permute the vertex IDs as the last step.

Note that since selection-inference consists of two subtasks, we have to encode the inputs differently. Rather than a list of visited vertices, we encode the list of visited edges. The example in Figure 13 would look like:

Input: **E** 4 1 **E** 8 3 **E** 3 6 **E** 8 4 **E** 2 3 **Q** 8 6 **P P P P P P** 8 4 **P** 4 1 **P**,  Label: 8

Input: **E** 4 1 **E** 8 3 **E** 3 6 **E** 8 4 **E** 2 3 **Q** 8 6 **P P P P P** 8 4 **P** 4 1 **P** 8,  Label: 3

The top example is one of the *selection* subtask, where the model must predict a previously-visited vertex with unvisited child vertices. The bottom is an example of the *inference* subtask, where given the vertex 8, the model must predict an unvisited child vertex. Interestingly, we find transformers do well on the selection subtask, but fare poorly on the inference subtask when given large input graphs.

## A.11 SELECTION-INFERENCE SCALING

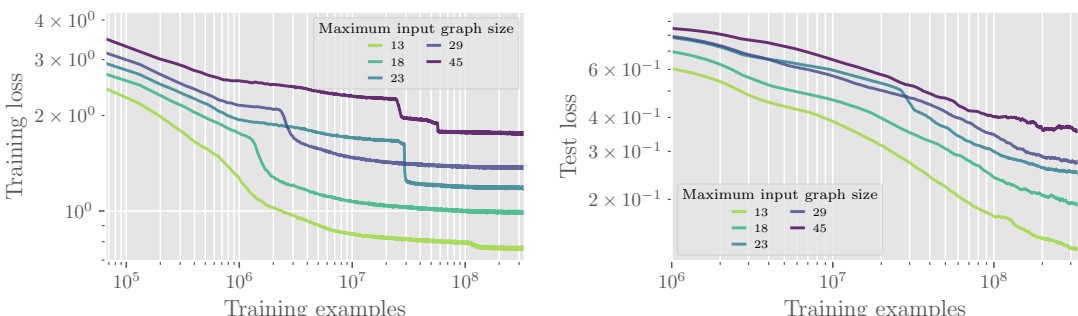

FIGURE 16: Training and test loss vs number of training examples seen, for models trained on the selection-inference task. Test loss is computed on held-out examples from the balanced distribution with frontier size 4 and branch count 4. We fix the model size and vary the maximum input graph size. All models were trained on the balanced distribution. Test loss is smoothed by averaging over a window of 81 data points, where each data point is recorded at every $2^{18} = 262\text{K}$ examples.

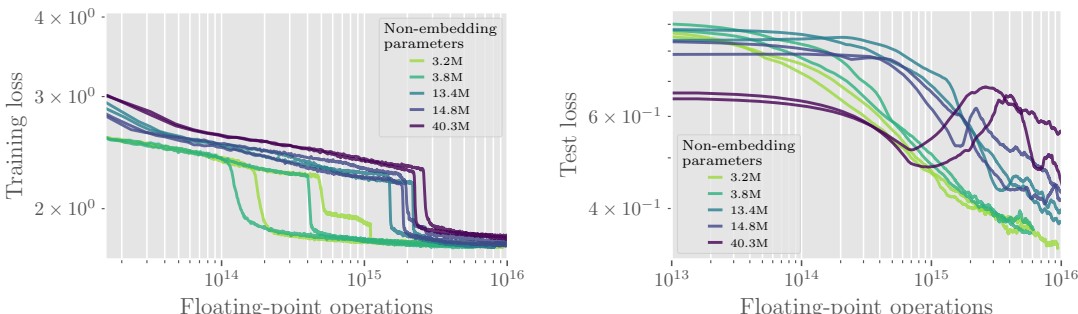

FIGURE 17: Training and test loss vs FLOPs, for models trained on the selection-inference task. Test loss is computed on held-out examples from the balanced distribution with frontier size 4 and branch count 4. We fix the maximum input graph size to 45 vertices and vary the model size. All models were trained on the balanced distribution. Test loss is smoothed by averaging over a window of 81 data points, where each data point is recorded at every $2^{18} = 262\text{K}$ examples.

## A.12 DOES THE TRANSFORMER SEARCH FORWARDS OR BACKWARDS?

We investigate whether the trained model performs search forwards from the start vertex or backwards from the goal vertex. To do so, we perform our mechanistic interpretability analysis on the trained model, using 100 examples of graphs with lookahead 9. We then restrict our attention to the vertices along the path from the start to the goal vertex, and count the number of path-merge operations between vertices along this path. We perform this analysis for two trained models: (1) the model trained on the balanced distribution with maximum lookahead 20, and (2) the model trained on the balanced distribution with maximum lookahead 12. Figure 18 visualizes the distribution of path-merge operations for the first model, and Figure 19 visualizes the same for the second model.

In the first model, we see a clear pattern in the first 4 layers where each target vertex is copying information from earlier and earlier source vertices (vertices that are progressively closer to the start vertex). Specifically, in layers 1 and 2, the $i^{th}$ vertex along the path from the start to the goal vertex is predominantly copying from the $i - 1^{th}$ vertex along the path. In layer 3, the $i^{th}$ vertex is copying from the $i - 2^{th}$ vertex. In layer 4, the $i^{th}$ vertex is copying from the $i - 4^{th}$ vertex. Thus, this particular model seems to be performing the search backwards from the goal.

However, in the second model (trained with maximum likelihood 12), the pattern is less clear. Layers 1 and 4 suggest the search is performed backwards, similar to the first model, since most path-merge operations lie above the main diagonal. However, layers 2 and 3 suggest the search is being performed forwards, as most of the path-merge operations lie below the main diagonal. We conclude that the search direction may be dependent on the random initialization seed, with some trained

models performing search forwards and others performing search backwards. Further experiments are necessary to obtain a more complete description.

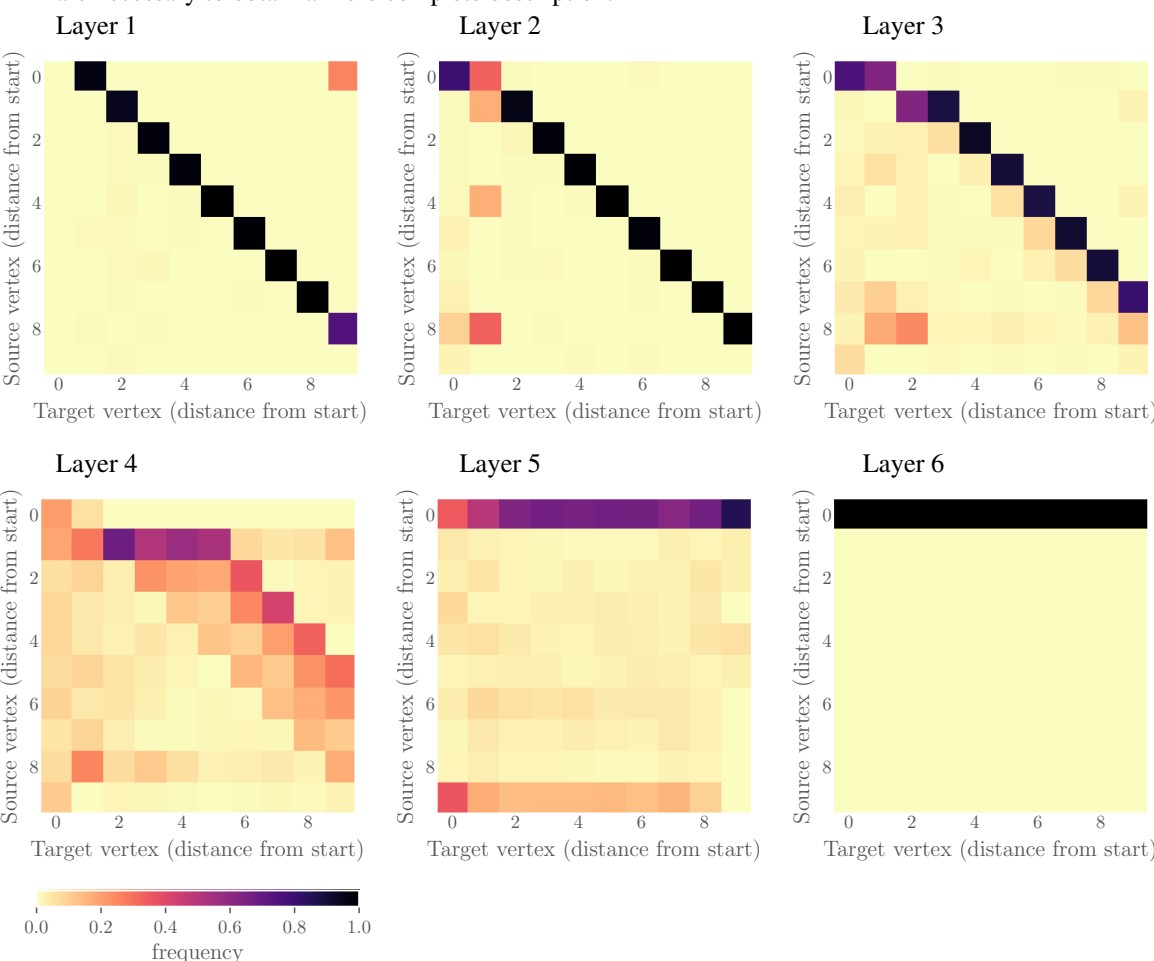

FIGURE 18: Visualization of the distribution of path-merge operations at each layer of the model trained on the balanced distribution with maximum lookahead 20. 100 input examples with lookahead 9 were analyzed to obtain the set of path-merge operations. We restrict the visualization to path-merge operations between vertices along the path from the start vertex to the goal vertex. The cell at row $i$ and column $j$ depicts the number of path-merge operations where the source is the $i^{th}$ vertex on the path and the target is the $j^{th}$ vertex on the path, divided by the total number of path-merge operations whose target is the $j^{th}$ vertex on the path.

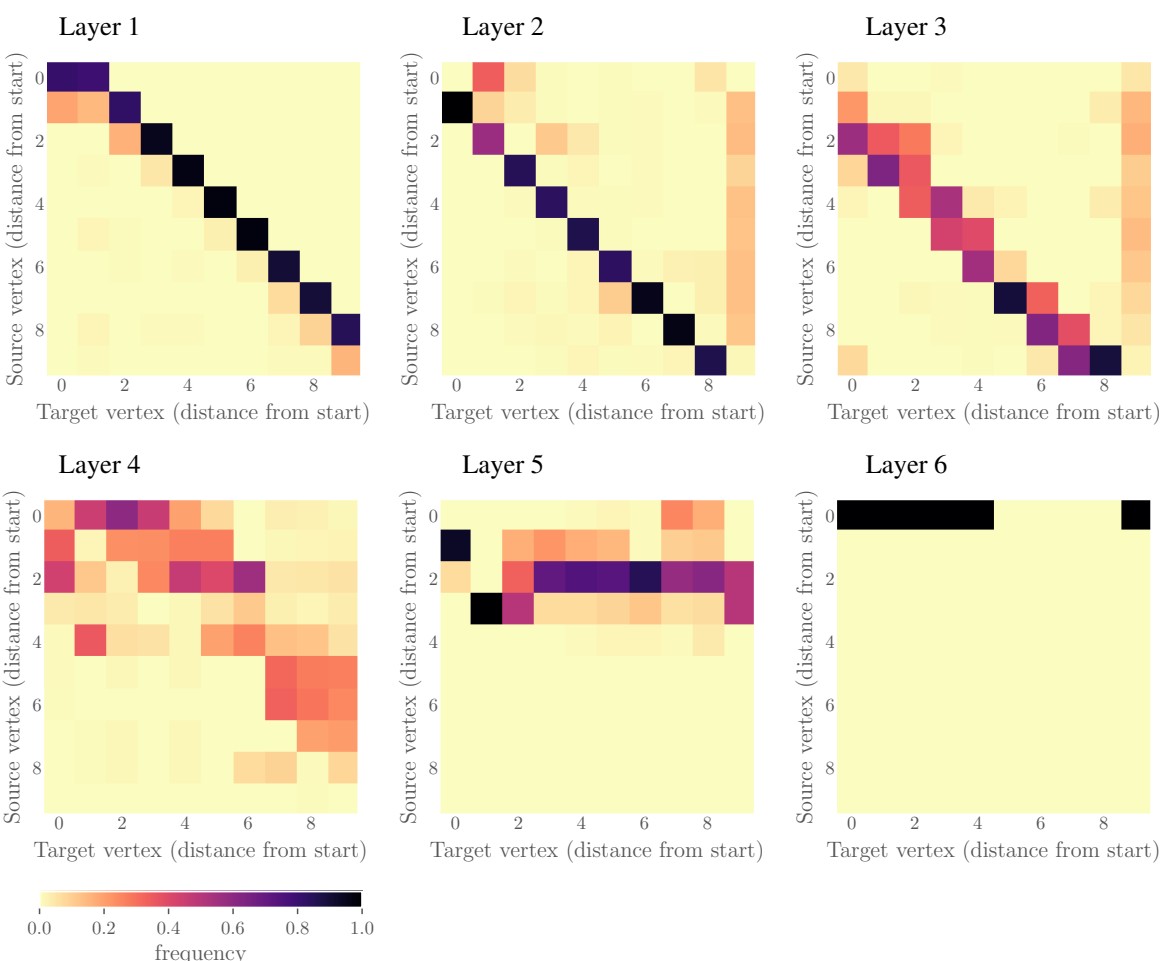

FIGURE 19: Visualization of the distribution of path-merge operations at each layer of the model trained on the balanced distribution with maximum lookahead 12. 100 input examples with lookahead 9 were analyzed to obtain the set of path-merge operations. We restrict the visualization to path-merge operations between vertices along the path from the start vertex to the goal vertex. The cell at row $i$ and column $j$ depicts the number of path-merge operations where the source is the $i^{th}$ vertex on the path and the target is the $j^{th}$ vertex on the path, divided by the total number of path-merge operations whose target is the $j^{th}$ vertex on the path.

