# OpenReview forum: "Transformers Struggle to Learn to Search"
_ICLR.cc/2025/Conference — ICLR 2025 Poster_

### Official Review · Reviewer_wdgg · 2024-11-02

**Soundness:** 3
**Presentation:** 3
**Contribution:** 3
**Rating:** 8
**Confidence:** 3

**Summary:**

This paper explores the ingredients that constitute the search ability in pre-trained language models. The authors introduce a synthetic setting--searching over DAGs represented in the natural language space--and pre-train autoregressive transformers of varying scale. The finding is mixed: transformers can implement search under restrictive data distribution, but face significant challenges with scaled problem sizes.  The authors have explored training strategies that encourage the transformer to generalize better.

**Strengths:**

- Combining mechanistic interpretability with the search problem is, to my knowledge, novel, as most prior works have focused on "classification"-style tasks that feature a very restricted output space in terms of vocabulary and length. The problem is more challenging in term of complexity, and would presumably require chain-of-thought capabilities to solve effectively. This new setting also prompts the author to introduce a new algorithm for mechanistic analysis, which may be of interest to the interpretability community.
- I enjoyed the exposition style presentation of the paper, with each section introducing problem setting of growing complexity, as well as experimental findings that sufficiently supports these findings.
- The authors study nuanced challenges for transformers to generalize on algorithmic tasks in the presence of distribution shift.

**Weaknesses:**

- My primary concern of this paper stems from the broader implication of the authors findings. Several prior works have found that large-language models can implement certain graph algorithms [1][2], including graph connectivity, and that this type of algorithmic reasoning can be improved with appropriate adaptations of chain-of-thought [3]. It is unclear whether the authors findings contradict, confirm, or offer more nuanced insights to prior works.
- While the paper does a fine job surveying relevant works in mechanistic interpretability, it is somewhat lacking when situating itself in the LLM planning/search and theoretical expressivity literature. Aside from the aforementioned works, several works have directly studied whether LLM can internalize search (in the form of MCTS) [4] and explore in-context [5]. The lack of a theoretical analysis, or a proper discussion of them make understanding the authors' contribution challenging.
- While the strategy that strengthens the LLM's search ability in the means of data augmentation is nice, it may not directly translate to practical guidance due to the synthetic nature of the task setup.

Overall, this is an interesting paper in terms of its mechanistic study; but I would encourage the author to situate its theses more broadly with the recent growing body of work in LLM exploration, search, and algorithmic reasoning.

[1] NPHardEval: Dynamic Benchmark on Reasoning Ability of Large Language Models via Complexity Classes, https://arxiv.org/abs/2312.14890

[2] IsoBench: Benchmarking Multimodal Foundation Models on Isomorphic Representations, https://arxiv.org/abs/2404.01266

[3] The Expressive Power of Transformers with Chain of Thought, https://arxiv.org/abs/2310.07923

[4] Amortized Planning with Large-Scale Transformers: A Case Study on Chess, https://arxiv.org/abs/2402.04494

[5] Can large language models explore in-context?, https://arxiv.org/abs/2403.15371

**Questions:**

See weakness.

---

> ### Author Response · Authors · 2024-11-23
> **Response to Reviewer**
>
> We thank the reviewer for their thoughtful and helpful feedback. We would greatly appreciate if the reviewers could inform us whether the revisions are sufficient or if there are any additional concerns, so that we may further iterate during the revision period.
>
> 1. “My primary concern of this paper stems from the broader implication of the authors findings. Several prior works have found that large-language models can implement certain graph algorithms [1][2], including graph connectivity, and that this type of algorithmic reasoning can be improved with appropriate adaptations of chain-of-thought [3]. It is unclear whether the authors’ findings contradict, confirm, or offer more nuanced insights to prior works. While the paper does a fine job surveying relevant works in mechanistic interpretability, it is somewhat lacking when situating itself in the LLM planning/search and theoretical expressivity literature. Aside from the aforementioned works, several works have directly studied whether LLM can internalize search (in the form of MCTS) [4] and explore in-context [5]. The lack of a theoretical analysis, or a proper discussion of them make understanding the authors' contribution challenging.”
>
> We thank you for raising this concern. We agree that the paper would benefit from additional related work on the expressivity of transformers as well as prompting-based approaches for search. As such, we have added several sentences to the revised Related Work section and included numerous additional references, including those suggested by the reviewer (thank you!). Notably, we include a discussion of whether transformers can express search algorithms versus whether they can learn those algorithms from data.
>
> To address each of the reviewer’s specific points: [1] and [2] show that transformers are able to perform some graph reasoning, but their abilities are certainly imperfect, and the graph sizes they consider are much smaller than those that we generate. Additionally, the question that we aim to answer is whether better training and/or additional scale can help further improve transformer’s graph reasoning abilities. While [4] demonstrates that transformers can approximate classical search algorithms, they note that there exists a gap due to approximation, and they do not test whether this gap would be narrowed with further training/scale. [5] found that GPT-4 is only able to engage in exploration when given an “externally summarized interaction history” and that “all other configurations did not result in robust exploratory behavior, including those with chain-of-thought reasoning but unsummarized history.” [3] does indeed show that transformers, equipped with the ability to generate intermediate steps (such as in chain-of-thought), are sufficiently expressive to simulate any Turing machine. However, the expressiveness of a task is certainly not the same as its learnability by a transformer, and our empirical study aims to better understand the learnability of the graph search task by transformers.
>
> [1] NPHardEval: Dynamic Benchmark on Reasoning Ability of Large Language Models via Complexity Classes, https://arxiv.org/abs/2312.14890
>
> [2] IsoBench: Benchmarking Multimodal Foundation Models on Isomorphic Representations, https://arxiv.org/abs/2404.01266
>
> [3] The Expressive Power of Transformers with Chain of Thought, https://arxiv.org/abs/2310.07923
>
> [4] Amortized Planning with Large-Scale Transformers: A Case Study on Chess, https://arxiv.org/abs/2402.04494
>
> [5] Can large language models explore in-context?, https://arxiv.org/abs/2403.15371
>
>
> 2. “While the strategy that strengthens the LLM's search ability in the means of data augmentation is nice, it may not directly translate to practical guidance due to the synthetic nature of the task setup.”
>
> We agree that the current results do not necessarily lend themselves to practical guidance, and this was not a goal of our work, though it is an interesting direction for future work. We do experiment with a natural language proof search version of the task in Section 3.1.2, where the length of each sentence can vary. Additional work is needed to further translate the sentences into more naturalistic forms in order to hopefully demonstrate promise in more practical applications.

---

> > ### Comment · Reviewer_wdgg · 2024-11-25
> >
> > Thank you for your detailed response. I'm happy with the updated paragraph. I would give a paper an updated score that's close to 7. But since we do not have this option, I'm keeping the score as is, but nevertheless be happy to see this paper at ICLR.
> >
> > I encourage the authors to address other lingering concerns and questions from other reviewers.

---

> > > ### Author Response · Authors · 2024-11-26
> > >
> > > We thank the reviewer for their quick response. We welcome any other suggestions or comments to help further improve the paper during the rebuttal period.

---

> > > > ### Comment · Reviewer_wdgg · 2024-12-02
> > > >
> > > > I have read the authors' rebuttals to other reviewers, and I believe the additional experiments have further strengthened this paper. I have adjusted my score accordingly.

---

### Official Review · Reviewer_zLV2 · 2024-11-04

**Soundness:** 3
**Presentation:** 3
**Contribution:** 3
**Rating:** 8
**Confidence:** 3

**Summary:**

The paper explores the behavior of transformers models when trained on search questions on directed acyclic graphs (DAGs). The authors show the importance of training data distribution for better generalization. Then, they conduct mechanistic understanding of the trained models to discover a progressive message passing algorithm utilized by the model to explore search paths. However, the models struggle to learn from larger graphs. Finally, the authors propose proxy in-context examples that help the model for robust exploration of the graph before solving the search problem. Overall, the paper represents a significant step towards our understanding of the inner mechanisms of transformer models.

**Strengths:**

The major strength of the paper lies in its motivation to understand transformer mechanisms in search based tasks. The authors take carefully designed experimental exploration to train transformers on directed acyclic graphs, with careful design discussion on data distribution, and propose a new mechanistic approach to analyze the learned algorithm. The authors discover a message passing algorithm, where the neighborhood information are shared progressively among the vertices, which leads to an exponential path-merging algorithm.

The authors also touch upon the difficulty of training transformer models on larger graph structures, and propose in-context tasks to help the model explore the graph better. Overall, the paper conducts an in-depth analysis of transformer model training on search problems and will be an important contribution to the academic community.

**Weaknesses:**

As such, the paper doesn't have many weaknesses. I have a couple of questions regarding the experimental setup.

a) **Sequence length in In-context exploration:** As the experiments require training on higher sequence length, how are the samples in training data distribution decided? How many steps in DFS traces are necessary for the model to learn? If the authors had provided same 'K' padding tokens to the experiments in the experiments in section 4, would the models generalize better?

b) **Distribution of path-merge operations:** Are there patterns in the distribution of path-merge operations and copy operations across the layers in the trained model?

c) **Evaluation with density of graphs?:** Do the trained models generalize to extremely sparse graphs?
- Furthermore, on cases where the graph contains $2$ disconnected components, what will the model output be for start and goal vertices not in the same component?

d) **Values of $\alpha$, $\kappa_1$ and $\kappa_2$ in section 4**: How are these values decided in experiments?

e) **Clarification questions:**

- "the log attention weight of each important operation in the last layer." (line 303) -
What does log attention weight mean? How do you define important operation?
- "it requires many forward passes (linear in the number of attention operations and in the number of input examples)." (line 354) -  Can the authors give details on the number of passes necessary? Furthermore, do the number of necessary passes depend (logarithmically) on the length of the search process for a given input example?

**Questions:**

Please check my questions in the above section.

---

> ### Author Response · Authors · 2024-11-23
> **Response to Reviewer (1/2)**
>
> We thank the reviewer for their thoughtful and helpful feedback. We would greatly appreciate if the reviewers could inform us whether the revisions are sufficient or if there are any additional concerns, so that we may further iterate during the revision period.
>
> 1. As the experiments require training on higher sequence length, how are the samples in training data distribution decided?
>
> When training on the balanced distribution, we first sample the desired backtrack distance uniformly at random from {1,...,B_max} where B_max is the largest possible backtrack for any graph that can fit in the transformer’s input. Next, we generate a graph with the selected backtrack, and repeat until we have generated a full batch (this is analogous to how we generate examples with uniform lookahead in Section 3/4). Thus, we generate graphs with a wide variety of sizes and backtrack distances. Please refer to Appendix Section A.7 for further details on the precise generative process.
>
>
> 2. How many steps in DFS traces are necessary for the model to learn?
>
> In our experiments, we show examples with DFS traces of many different lengths. This is required if using standard padding since the transformer would not be able to generalize to DFS traces that are longer than those seen during training. However, if you use random padding, the model can be taught to perform DFS using only examples with shorter traces. Though further experimentation would be interesting to explore this further.
>
>
> 3. If the authors had provided same 'K' padding tokens to the experiments in section 4, would the models generalize better?
>
> We don’t think so, since the experiments in Section 4 were on a task that’s more akin “direct prompting.” In this task the model is only given the start vertex and asked to predict the next vertex on the path to the goal vertex. As such, the model is required to perform the full search in a single forward pass. Padding is unnecessary since the graph edges already appear in the same input positions across examples in the Section 4 experiments.
>
> 4. Do the trained models generalize to extremely sparse graphs? Furthermore, on cases where the graph contains disconnected components, what will the model output be for start and goal vertices not in the same component?
>
> We only generate connected graphs. Though we do generate connected graphs with minimal edges (such as those with maximal lookahead) and we find that if the model is able to reach 100% training accuracy, it can correctly perform search on these minimal connected graphs. However, we have not experimented with graphs with multiple components, and we agree this would be interesting to explore.
>
> But if a model trained on connected graphs were given a disconnected graph where the goal is unreachable, then we suspect the model would randomly predict one of the child vertices of the start vertex, since this heuristic would work reasonably on the training examples.
>
>
> 5. Values of $\alpha$, $\kappa_1$ and $\kappa_2$ in section 4: How are these values decided in experiments?
>
> These three parameters determine the sensitivity of the mechanistic interpretability analysis, where if they are set loosely, the analysis will find that many more attention operations are “important”. But this increases the computational cost of the analysis. If the parameters are set too strictly, the analysis may miss some important attention operations and fail to reconstruct the computation graph. Thus, there is a tradeoff between sensitivity and computational cost, and we selected the values to be loose enough to identify the path-merging algorithm for most inputs, but not much looser. We also demonstrate the analysis is highly specific as we apply it to an untrained transformer model (random weights) in Figure 6.
>
>
> 6. "the log attention weight of each important operation in the last layer." (line 303) - What does log attention weight mean? How do you define important operation?
>
> This is the attention weight _before_ the softmax operation. An attention operation is defined to be _important_ if perturbing the weight causes a sufficiently large change in the output logits. We describe how we compute this in Steps 2 and 3 of Section 4.1.

---

> > ### Author Response · Authors · 2024-11-23
> > **Response to Reviewer (2/2)**
> >
> > 8. "...It requires many forward passes (linear in the number of attention operations and in the number of input examples)." (line 354) - Can the authors give details on the number of passes necessary? Furthermore, do the number of necessary passes depend (logarithmically) on the length of the search process for a given input example?
> >
> > The number of forward passes is $L n^2 m F$ where $L$ is the number of layers, $n$ is the input length of the transformer, $m$ is the number of input examples, and $F$ is the number of perturbed features. We have added a footnote to this sentence in the revised submission for clarification. We also note that the method as presented is not practically applicable to very large models, but we do demonstrate its applicability and utility to our trained models.
> >
> > The number of forward passes does not directly depend on the length of the search process. However, since we show that the minimum number of layers needed to perform search with lookahead $L$ is ~$\text{log}_2(L)$, and so there could be an indirect relation if the number of layers was specifically selected for a particular target lookahead.

---

> ### Comment · Reviewer_zLV2 · 2024-12-02
>
> I thank the authors for their detailed responses. I believe this is a strong paper, hence I am increasing my score to 8.

---

### Official Review · Reviewer_fM1a · 2024-11-07

**Soundness:** 2
**Presentation:** 3
**Contribution:** 3
**Rating:** 6
**Confidence:** 3

**Summary:**

This paper aims to explore LLMs' internal mechanisms for graph connectivity problems tasks: given a graph (nodes and connections between nodes), a starting vertex, and a goal vertex, the LLM outputs the next vertex from the starting vertex.
Specifically, the paper constructs a training set to train a small decoder-only transformer. It improves the Mechanistic Interpretation visualization method to explore which tokens influence the LLM's output. Based on experimental results, the authors conclude that LLMs must be trained with in-context samples to fully understand the graph search problem.

**Strengths:**

The topic is interesting and may have influence in the community.

**Weaknesses:**

1. There are potential data leakage issues in the training and testing datasets constructed by the authors:
The authors use a generation method to generate training data online and save the first few generated results as test data. While the authors claim they will remove overlapping samples between training and test data, they don't explain how they compare whether two graphs are identical. If only using string matching, it cannot determine whether two graphs are completely equal. For example, these two graphs: node 1 -> node 2 and node 3 -> node 4 are completely equivalent but cannot be detected through string matching. Therefore, the test set is likely included in the training set. Additionally, given the number of vertices and max number of in-out edges, DAG generation is finite. Thus, the authors' claim about infinite graph generation may be incorrect.
The authors trained a simplified model that doesn't correspond to currently widely-used LLMs:
First, since the authors used full attention rather than causal attention, they actually trained an encoder-only model rather than a decoder-only model.

2. Second, the authors only used one-hot embedding for each token and position embedding when training this encoder-only model.
Finally, for the graph connectivity problem, the authors' trained model only outputs one token, while current LLMs typically have reasoning steps.

3. The authors' improved Mechanistic Interpretation has the following issues:
The proposed method requires performing perturbation and forward pass once for each element in every attention map of the LLM, and then forward pass to see the effects on the output logits, which is time consuming.
In Line 318, determining the influence of modified tokens on attention through frozen previous layers is not reasonable, as modified tokens also influence previous attention calculation and thus influence the activation for each layer.

**Questions:**

Please refer to the weaknesses part.

---

> ### Author Response · Authors · 2024-11-23
> **Response to Reviewer (1/2)**
>
> We thank the reviewer for their thoughtful and helpful feedback. We would greatly appreciate if the reviewers could inform us whether the revisions are sufficient or if there are any additional concerns, so that we may further iterate during the revision period. We clarify that the focus of our analysis is on small transformer models, rather than current large language models. We provide the transformer with effectively limitless and idealized examples, with the aim being to estimate an “upper bound” on the model’s ability to learn to search. We have added sentences and modified the language in the introduction to better reflect our high-level aims.
>
> 1. “There are potential data leakage issues in the training and testing datasets constructed by the authors: The authors use a generation method to generate training data online and save the first few generated results as test data. While the authors claim they will remove overlapping samples between training and test data, they don't explain how they compare whether two graphs are identical.”
>
> We use exact string matching to filter examples from the training set that appear in the test set. While it is true that this would not identify graphs that are isomorphic (which would be computationally intractable to compute), we do not _need_ to fully control for data leakage to demonstrate our claims. In Sections 5 and 6, we perform scaling experiments whose goal is to effectively compute an _upper bound_ on the transformer’s ability to learn to search. We provide the model with effectively limitless and idealized training data, which has been carefully curated to preclude the learning of heuristics or shortcuts. However, if the transformer does end up memorizing some of the data, or learning a heuristic, the measured performance would only increase, and our measured upper bound is still valid.
>
> We are not aware of any work that suggests that transformers can generalize to isomorphic graphs. Our mechanistic interpretability analysis also shows that the transformer is indeed learning a robust and generalizable algorithm that explains its behavior on almost all inputs. Furthermore, in the last row of Figure 3, we observe that the model trained on graphs with lookaheads up to 12 is able to generalize to lookaheads 13 and 14, which would not be possible if the model has memorized the graph topologies of a large portion of training examples, since there are no graphs with lookaheads larger than 12 in the training distribution.
>
> In order to more clearly convey our high-level aim, we add the following sentence to the second paragraph in the Introduction:
>  > By automatically generating such examples, we provide the transformer with effectively limitless and idealized training data, with which we can estimate an ``upper bound'' on the transformer's ability to learn to search.
>
> We additionally rephrase the sentence in the second paragraph of 3.1 to clarify the filtering procedure:
>  > The first few batches are reserved for testing, and subsequent batches are filtered via exact matching to remove any examples that appear in the test set, to ensure that the examples in the test set are unseen.
>
>
> 2. “Given the number of vertices and max number of in-out edges, DAG generation is finite. Thus, the authors' claim about infinite graph generation may be incorrect.”
>
> We say the amount of training data is “effectively limitless” only to draw comparison with the non-synthetic datasets which have a fixed size. In contrast, with sufficiently many vertices, the number of possible DAGs is quite high (see https://oeis.org/A003024), and we are limited entirely by compute rather than by data availability.
>
> In addition, since transformers have a fixed input size (and limited precision), it is impossible to provide more than a finite number of graphs as input to the transformer. So the more precise question that we seek to answer is not whether transformers can learn to search on _all_ graphs, but whether they can learn to search on any graph that can be provided as input.
>
>
> 3. “The authors only used one-hot embedding for each token and position embedding when training this encoder-only model.”
>
> The purpose of this was to facilitate the mechanistic interpretability analysis in Section 4.

---

> > ### Author Response · Authors · 2024-11-23
> > **Response to Reviewer (2/2)**
> >
> > 4. “For the graph connectivity problem, the authors' trained model only outputs one token, while current LLMs typically have reasoning steps.”
> >
> > We tested the effect of intermediate steps in Section 6. Here, rather than teaching the model to perform the full search in a single forward pass, we instead train the model to predict the next step in a depth-first search (DFS). Thus, in this task, the transformer only needs to predict the next step (which may or may not lead to the goal). We added new results on the scaling behavior of transformers on this DFS task and we found that while they are able to learn the task more easily, they still struggle on larger graphs (see revised Section 6 and Figure 12).
> >
> > In addition, we also test the scaling behavior of transformers on the selection-inference task [1], where each step in the search is broken into two subtasks: (1) select a previously-visited vertex with unvisited child vertices, and (2) from the current vertex, select an unvisited child vertex. We find that transformers are able to learn the first subtask (i.e., “selection”) relatively easily, they struggle with the second subtask (i.e., “inference”) when given larger graphs (see Section 6.2, and Figures 14 and 15).
> >
> > [1] Antonia Creswell, Murray Shanahan, Irina Higgins: Selection-Inference: Exploiting Large Language Models for Interpretable Logical Reasoning. ICLR 2023.
> >
> >
> > 5. “The proposed method requires performing perturbation and forward pass once for each element in every attention map of the LLM, and then forward pass to see the effects on the output logits, which is time consuming.”
> >
> > We acknowledge that our proposed analysis is computationally expensive. However, we do not aim to apply our method to current LLMs. Rather, we only apply our method to small transformer models which we train in our experiments, and we demonstrate the utility of our method on such models in Section 4.2 by performing our analysis on 2,000 input examples _each_ on 4 different models.
> >
> >
> > 6. “In Line 318, determining the influence of modified tokens on attention through frozen previous layers is not reasonable, as modified tokens also influence previous attention calculation and thus influence the activation for each layer.”
> >
> > The goal of this step is specifically to _isolate_ the effect of the perturbed tokens on the attention operation being studied. We do this after performing the very same step on the attention operations in the previous layers, and so by this point in the analysis, we have already characterized the influence of perturbations on the attention of all previous layers. To hopefully clarify this in the text, we add the following sentence to Footnote 3:
> >  > The aim of freezing the previous layers is to measure the effect of the perturbation on the current layer _in isolation_ of changes in behavior in preceding layers.

---

> > > ### Comment · Reviewer_fM1a · 2024-11-26
> > > **Response to the rebuttal**
> > >
> > > I appreciate the authors' rebuttal and their investigation into LLMs' capabilities for graph search problems. While the work presents an interesting analytical approach, I still have some major concerns about the experimental setup that I believe greatly weaken its impact:
> > >
> > > 1. The choice of an encoder-only architecture, while valid for analysis, may limit the generalizability of the findings to contemporary LLMs, which predominantly use decoder-only architectures. I appreciate that the authors have revised their description from "decoder-only transformers" to simply "transformers" - a notable correction given that their original manuscript described using decoder-only models with bidirectional attention masks. As this represents a basic architectural contradiction in transformer design (decoder-only models, by definition, use causal masking, not bidirectional attention).
> > >
> > > 2. The simplification of using one-hot embeddings with concatenated positional embeddings might not capture the complexity of modern embedding approaches.
> > >
> > > 3. The assumption about token modifications affecting only subsequent layers may oversimplify the intricate dynamics of attention mechanisms, as modifications can propagate through the entire network.
> > >
> > > 4. Given the authors' acknowledgment that the proposed analysis may not be directly applicable to current LLMs due to computational constraints, it would be valuable to elaborate on the broader implications and potential future applications of this research direction.

---

> ### Author Response · Authors · 2024-11-28
> **Response to Reviewer**
>
> We thank the reviewer for providing additional feedback, and we welcome further discussion so that we can continue to further improve the submission.
>
> 1. On the point on the encoder-only architecture and simplified positional embeddings:
>
> We agree with the reviewer that contemporary LLMs are largely implemented with decoder-only transformers and dense positional embeddings. As such, we have re-run our scaling experiments in Section 5 with decoder-only transformers as well as learnable token embedding and rotary positional embeddings (summed rather than concatenated). We add these results to Section A.5 (see Figures 11 and 12) in the new revision. We find that the causal mask and rotary positional embeddings do not yield a significant difference in the model's scaling behavior on the search task.
>
> 2. On elaborating on the broader implications and potential future applications of the proposed mechanistic interpretability analysis:
>
> The conclusion contains some sentences discussing the broader implications, but we have the following to elaborate further:
>  > Though additional work is welcome to improve the scalability of our analysis to larger models, our analysis can provide insights on smaller models that can be tested separately in larger models.
>
> 3. On the point about the effect of token perturbations in the mechanistic interpretability analysis:
>
> We do not make the assumption that token perturbations only affect subsequent layers. To clarify, we start our analysis with the first layer, performing a forward pass (only up to the first attention block) on both the original and perturbed inputs. Then for each element in the attention matrix, we compute the product of the original key vector with the corresponding _perturbed_ query vector (and similarly, we compute the dot product of the perturbed key vector with the corresponding original query vector). By observing the change in the perturbed dot products relative to the original dot product, we characterize the dependence of each attention operation on the input features. Once this is done for the first layer, we move onto the second layer and repeat the analysis, and so we have already obtained a mechanistic description of the first layer's behavior without any assumptions on the effect of the perturbations on subsequent layers.

---

> ### Comment · Reviewer_wdgg · 2024-12-02
> **Additional Comments on the Author Response**
>
> Dear Reviewer fM1a,
>
> As a fellow reviewer, I hope to address some of your followup questions to contextualize some of the authors' design choices.
>
> ### **Encoder vs. Decoder**
>
> Most of the existing studies on mechanistic interpretability (in particular those pertaining to training dynamics) uses some simplified transformer variants. For example, [1] proposes an architecture that circumvents the additive structure of the residual stream, [2][3] use linear attention with linear attention. While it's desirable to study an architecture that mimics LLMs, the current progress should allow for mediated design choices.
>
> ### **Concatenated Embedding**
>
> Again this seems to be a common assumption that theoretical transformer papers tend to make (see e.g. [4][5]), and it can often be shown WLOG that this concatenation can be converted to the typical additive structure with 1 additional layer.
>
> Since we still understand very little about transformer interpretability, I think that it's reasonable to operate with mediated expectations and study stylized problems.
>
> [1] How Transformers Learn Causal Structure with Gradient Descent, https://arxiv.org/abs/2402.14735
>
> [2] Birth of a Transformer: A Memory Viewpoint, https://arxiv.org/abs/2306.00802
>
> [3] Progress measures for grokking via mechanistic interpretability, https://arxiv.org/abs/2301.05217
>
> [4] Transformers Learn to Achieve Second-Order Convergence Rates for In-Context Linear Regression, https://arxiv.org/abs/2310.17086
>
> [5] Transformers as Statisticians: Provable In-Context Learning with In-Context Algorithm Selection, https://arxiv.org/abs/2306.04637

---

> > ### Comment · Reviewer_fM1a · 2024-12-02
> > **Response to the rebuttal**
> >
> > I thank the authors and Reviewer wdgg for their comprehensive response. Since the settings align with common practices in transformer theoretical analysis, and given the additional experiments provided by the authors, I have raised my score accordingly. I strongly recommend including more detailed descriptions of the supplementary experiments on decoder-only models to enhance the paper's broader impact.

---

### Official Review · Reviewer_NMwf · 2024-11-11

**Soundness:** 2
**Presentation:** 1
**Contribution:** 2
**Rating:** 5
**Confidence:** 3

**Summary:**

This study investigates whether transformers can learn to perform search by training small models on graph connectivity data. Results show that transformers can learn to search under certain conditions, but struggle with larger graphs, indicating that simply scaling LLMs may not enable robust search. The study introduces a new interpretability method to analyze the model's learned algorithm.

**Strengths:**

- The study tackles an intriguing and practical research question: understanding the mechanisms behind search capabilities in LLMs. This is not only scientifically interesting but also has meaningful implications for real-world applications.
- Training a small GPT model on synthetic graph data is a reasonable and well-justified approach to investigate this research question.

**Weaknesses:**

The logical flow of the paper is weak in several areas. The authors should clarify the connection between their empirical results and the statements made, as well as provide more intuition behind their hypotheses.
For example, in line 51, the authors state, "We demonstrate experimentally that transformers can indeed be taught to search, but only under fairly restrictive conditions on the training distribution." However, Figure 3 does not fully support this claim. While it may indicate that the model does not generalize well to a larger number of lookaheads than seen in the training data, it does not substantiate any firm conclusions about the training distribution itself.
In line 359, the authors mention, "We noticed a pattern and formed a hypothesis about the algorithm the model has acquired to solve the search problem." However, the pattern observed and its connection to the proposed hypothesis remain unclear and should be elaborated upon.

Additionally, the proposed method and analysis require clarification. For instance, in line 337, the phrase "path of explainable attention operations" is used—was this path inspected manually? And in line 358, the authors mention "a number of input examples" without specifying the exact number. Providing this detail would help improve the robustness of their claim.

**Questions:**

- Interpretation of Figure 3: The paper claims that transformers can search under restrictive training distributions, but Figure 3 only seems to show limited generalization to larger lookaheads. Can the authors explain how this supports claims about the training distribution?
- Pattern and Hypothesis Formation (Line 359): What specific pattern did the authors observe, and how did it lead to the hypothesis about the algorithm the model uses? Could they provide a clear link between the observed pattern and their hypothesis?
- Explainable Attention Path (Line 337): What exactly is meant by a "path of explainable attention operations"? Was this path derived through manual inspection, or was there a specific method used?
- Quantifying Examples (Line 358): The authors mention using "a number of input examples" but do not specify the exact number. Could they provide this detail to strengthen the robustness of their conclusions?

---

> ### Author Response · Authors · 2024-11-23
> **Response to Reviewer**
>
> We thank the reviewer for their thoughtful and helpful feedback. We would greatly appreciate if the reviewers could inform us whether the revisions are sufficient or if there are any additional concerns, so that we may further iterate during the revision period.
>
> 1. “The authors should clarify the connection between their empirical results and the statements made, as well as provide more intuition behind their hypotheses. For example, in line 51, the authors state, `We demonstrate experimentally that transformers can indeed be taught to search, but only under fairly restrictive conditions on the training distribution.’ However, Figure 3 does not fully support this claim.”
>
> We agree with the reviewer that the clarity of the paper can be improved. On this specific comment, we rephrased the claim to more accurately convey what is shown in the results. Line 51 has been rephrased to
>  > We demonstrate experimentally that transformers can indeed be taught to search, when given the right training distribution.
>
> We likewise modified all similar claims made elsewhere in the paper, such as in the abstract. We make this point to contrast with recent work showing that transformers are not able to learn to search [1,2].
>
> To better support this rephrased claim, we add additional results to Figure 3, where we train and test models on an additional “star graph” distribution. We added a paragraph describing this distribution in Section 3, and updated the text in Figure 3 and Section 3.1.1. accordingly.
>
> [1] Honghua Zhang, Liunian Harold Li, Tao Meng, Kai-Wei Chang, Guy Van den Broeck: On the Paradox of Learning to Reason from Data. IJCAI 2023.
>
> [2] Gregor Bachmann, Vaishnavh Nagarajan: The Pitfalls of Next-Token Prediction. ICML 2024.
>
>
> 2. “In line 359, the authors mention, ‘We noticed a pattern and formed a hypothesis about the algorithm the model has acquired to solve the search problem.’ However, the pattern observed and its connection to the proposed hypothesis remain unclear and should be elaborated upon… And in line 358, the authors mention ‘a number of input examples’ without specifying the exact number.”
>
> We removed this opaque sentence. It was only meant to briefly describe the exploratory analysis that we conducted before forming our hypothesis. The more rigorous experiments that test this hypothesis are described in the next paragraph, where we added additional details, such as the fact that we performed our analysis on a total of 2000 inputs from the naive and balanced distributions (100 for each lookahead).
>
>
> 3. “...In line 337, the phrase ‘path of explainable attention operations’ is used—was this path inspected manually?”
>
> We agree the current description is unclear. The explainable attention operations are computed automatically. To improve clarity, we rewrite this paragraph:
>  > Starting from the first layer, let $t_k$ be the token at position $k$ of the input. We say each input vector ``_explainably contains_'' information about the token value $t_k$ and position $k$. Next, we consider the attention operations in the first layer. Suppose an attention operation copies source token $i$ into target token $j$, and depends on the source token embedding containing features $f^S_1,\ldots,f^S_u$ and depends on the target token embedding containing features $f^T_1,\ldots,f^T_v$ to perform this operation (as computed in Step 4). We say this attention operation is _explainable_ if the embedding of token $i$ explainably contains all features $f^S_1,\ldots,f^S_u$, and the embedding of token $j$ explainably contains all features $f^T_1,\ldots,f^T_v$. If the attention operation is explainable, we say the output embedding of the target token $j$ explainably contains the union of the features: $f^S_1,\ldots,f^S_u,f^T_1,\ldots,f^T_v$. We repeat this for each successive layer, computing all explainable attention operations throughout the model. Pseudocode for this procedure is shown in Algorithm 1.
>
> We also add an algorithm in the Appendix with pseudocode describing how we compute the set of explainable attention operations.

---

> > ### Author Response · Authors · 2024-12-03
> > **Reminder to Reviewer**
> >
> > Dear Reviewer NMwf,
> >
> > We are sending a reminder that the rebuttal period is ending soon. If you have any additional feedback, we would really appreciate it if you would provide them as soon as convenient.
> >
> > Sincerely,
> > Submission11660 Authors

---

### Author Response · Authors · 2024-11-23
**General Response to Reviewers**

We sincerely thank all reviewers for their thoughtful feedback. We have taken your comments into careful consideration and worked to improve on the paper. We have improved the clarity throughout the paper and tightened many of the claims.

Additionally, we added more complete experimental results, notably in Section 6, where we experiment with in-context (i.e., step-by-step) methods for solving the task, akin to chain-of-thought. We added results on scaling experiments for the depth-first search task (Section 6.1; Figure 14). We also experimented with selection-inference “prompting,” where each step of the search is decomposed into two subtasks: (1) select a previously-visited vertex that has unvisited child vertices, and (2) from the current vertex, predict an unvisited child vertex. We added scaling experiment results for this selection-inference task (Section 6.2; Figures 16 and 17). We found that, while depth-first search and selection-inference is easier to learn than direct prompting, the transformer still struggles on larger input graphs, and we find that increasing model scale does not mitigate this difficulty.

---

> ### Author Response · Authors · 2024-12-02
> **Reminder to Reviewers**
>
> Thank you to the reviewers for their responses thusfar. We believe we have addressed _**all**_ of your concerns in our last reply. We would really appreciate if you could let us know whether there are any additional concerns before the end of the rebuttal period (11:59pm today AoE).

---

### Meta-Review · Area_Chair_hSuN · 2024-12-18

**Metareview:**

This paper investigates the ability of large language models (LLMs), specifically transformers, to perform search tasks using graph connectivity problems as a testbed. The paper includes experiments on directed acyclic graphs (DAGs) and introduces novel mechanistic interpretability tools for this problem,  revealing that transformers compute reachable vertex sets layer by layer. All the reviewers appreciated the paper's contribution in terms of methodology and empirical findings, and agree than the paper is likely to be of significant interest to the broader ICLR community. The main weakness noted by the reviewers pertained to presentation and writing. Many of those were resolved by a thorough rebuttal and during the discussion phase. Additionally, reviewer fM1a raised concerns about a potential data leakage, which seems to have been resolved by the authors' response.

Based on the reviews, rebuttal, and discussion, I recommend accepting the paper. The main reasons for this decision are:
* **Significant Contribution**: The paper addresses a fundamental problem in the field, and adds to our growing understansding of the capabilities and limitations of LLMs. The core findings of the paper (e.g. the mechanism by which the models perform search over graphs) are to the best of mine and the reviewers' knowledge, novel.
* **Thorough Experimentation**: The experiments are well-designed and provide strong evidence for the paper’s claims.
* **Clear Presentation**: The paper is well-written and clearly presents its findings, making it accessible to a broad audience.

Overall, I think this paper would make a good addition to ICLR. I anticipate it will garner significant interest and will be heavily discussed.

**Additional Comments On Reviewer Discussion:**

The authors provided a detailed rebuttal addressing the reviewers’ concerns. They clarified the mechanistic interpretability section and expanded the discussion on potential solutions to the challenges identified. The authors’ responses were well-received by the reviewers, who acknowledged the improvements and maintained their positive evaluations.

The discussion among reviewers was constructive, focusing on the paper’s contributions and the importance of the problem it addresses. There was a consensus that the paper provides valuable insights into the limitations of transformers and offers a solid foundation for future research in this area. The reviewers agreed that the paper’s strengths outweigh the minor issues identified.

After the discussion phase, only one reviewer (NMwf) had a below-acceptance score. However, that reviewer stopped engaging after posting their review and did not react to the authors' response to their review (nor to my requests for discussion), which as far as I can tell addressed their main concerns, so I have downweighed that score in my decision.

---

### Decision · Program_Chairs · 2025-01-22

Accept (Poster)